# Rapid epidemic expansion of the SARS-CoV-2 Omicron variant in southern Africa

Raquel Viana[1,50], Sikhulile Moyo[2,3,4,50], Daniel G. Amoako[5,50], Houriiyah Tegally[6,50], Cathrine Scheepers[5,7,50], Christian L. Althaus[8], Ugochukwu J. Anyaneji[6], Phillip A. Bester[9,10], Maciej F. Boni[11], Mohammed Chand[12], Wonderful T. Choga[3], Rachel Colquhoun[13], Michaela Davids[14], Koen Deforche[15], Deelan Doolabh[16], Louis du Plessis[17,18], Susan Engelbrecht[19], Josie Everatt[5], Jennifer Giandhari[6], Marta Giovanetti[20,21], Diana Hardie[16,22], Verity Hill[13], Nei-Yuan Hsiao[16,22,23], Arash Iranzadeh[24], Arshad Ismail[5], Charity Joseph[12], Rageema Joseph[16], Legodile Koopile[2], Sergei L. Kosakovsky Pond[25], Moritz U. G. Kraemer[17], Lesego Kuate-Lere[26], Oluwakemi Laguda-Akingba[27,28], Onalethatha Lesetedi-Mafoko[29], Richard J. Lessells[6], Shahin Lockman[2,30], Alexander G. Lucaci[25], Arisha Maharaj[6], Boitshoko Mahlangu[5], Tongai Maponga[19], Kamela Mahlakwane[19,31], Zinhle Makatini[32], Gert Marais[16,22], Dorcas Maruapula[2], Kereng Masupu[4], Mogomotsi Matshaba[4,33,34], Simnikiwe Mayaphi[35], Nokuzola Mbhele[16], Mpaphi B. Mbulawa[36], Adriano Mendes[14], Koleka Mlisana[37,38], Anele Mnguni[5], Thabo Mohale[5], Monika Moir[39], Kgomotso Moruisi[26], Mosepele Mosepele[4,40], Gerald Motsatsi[5], Modisa S. Motswaledi[4,41], Thongbotho Mphoyakgosi[36], Nokukhanya Msomi[42], Peter N. Mwangi[10,43], Yeshnee Naidoo[6], Noxolo Ntuli[5], Martin Nyaga[10,43], Lucier Olubayo[23,24], Sureshnee Pillay[6], Botshelo Radibe[2], Yajna Ramphal[6], Upasana Ramphal[6], James E. San[6], Lesley Scott[44], Roger Shapiro[2,30], Lavanya Singh[6], Pamela Smith-Lawrence[26], Wendy Stevens[44], Amy Strydom[14], Kathleen Subramoney[32], Naume Tebeila[5], Derek Tshiabuila[6], Joseph Tsui[17], Stephanie van Wyk[39], Steven Weaver[25], Constantinos K. Wibmer[5], Eduan Wilkinson[39], Nicole Wolter[5,45], Alexander E. Zarebski[17], Boitumelo Zuze[2], Dominique Goedhals[10,46], Wolfgang Preiser[19,31], Florette Treurnicht[32], Marietje Venter[14], Carolyn Williamson[16,22,23,47], Oliver G. Pybus[17], Jinal Bhiman[5,7], Allison Glass[1,48], Darren P. Martin[23,47], Andrew Rambaut[13], Simani Gaseitsiwe[2,3,51], Anne von Gottberg[5,45,51] & Tulio de Oliveira[6,39,49,51✉]

The SARS-CoV-2 epidemic in southern Africa has been characterized by three distinct waves. The first was associated with a mix of SARS-CoV-2 lineages, while the second and third waves were driven by the Beta (B.1.351) and Delta (B.1.617.2) variants, respectively[1–3]. In November 2021, genomic surveillance teams in South Africa and Botswana detected a new SARS-CoV-2 variant associated with a rapid resurgence of infections in Gauteng province, South Africa. Within three days of the first genome being uploaded, it was designated a variant of concern (Omicron, B.1.1.529) by the World Health Organization and, within three weeks, had been identified in 87 countries. The Omicron variant is exceptional for carrying over 30 mutations in the spike glycoprotein, which are predicted to influence antibody neutralization and spike function[4]. Here we describe the genomic profile and early transmission dynamics of Omicron, highlighting the rapid spread in regions with high levels of population immunity.

Since the onset of the COVID-19 pandemic in December 2019, variants of SARS-CoV-2 have emerged repeatedly. Some variants have spread worldwide and made major contributions to the cyclical infection waves that occur asynchronously in different regions. Between October and December 2020, the world witnessed the emergence of the first variants of concern (VOCs). These variants exhibited increased transmissibility and/or immune evasion properties that threatened global efforts to control the pandemic. Although the Alpha (B.1.1.7), Beta and Gamma VOCs[2,5] that emerged during this time disseminated globally and drove epidemic resurgences in many different countries, it was the highly transmissible Delta variant that subsequently displaced all of the other VOCs in most regions of the world[6]. During its spread, the Delta variant evolved into multiple sublineages[7], some of which demonstrated signs of having a growth advantage in certain locations[8], prompting speculation that the next VOC to drive a resurgence of infections would probably be derived from Delta. In October 2021, while Delta was continuing to exhibit high levels of transmission in the Northern Hemisphere, a large Delta wave was subsiding in southern Africa. The culmination of this wave coincided with the emergence of a new SARS-CoV-2 variant that, within days of its near-simultaneous discovery in four individuals

in Botswana, a traveller from South Africa in Hong Kong and 54 individuals in South Africa, was designated by the World Health Organization (WHO) as Omicron—the fifth VOC of SARS-CoV-2. Since then and the beginning of 2022, over 100,000 genomes of Omicron have been produced as Omicron has started to dominate SARS-CoV-2 infections in the world.

## Epidemic dynamics and detection of Omicron

The three distinct epidemic waves of SARS-CoV-2 experienced by southern African countries were each driven by different variants: the first between June and August 2020 by descendants of the B.1 lineage[1]; the second between November 2020 and February 2021 by the Beta VOC[2,9]; and the third between May and September 2021 by the Delta VOC[3], with an estimated 2–5% of third-wave cases in South Africa attributed to the C.1.2 lineage[10] (Fig. 1a). Serosurveys conducted before the Delta wave suggested high levels of exposure to SARS-CoV-2 (40–60%) in South Africa[11,12], and the estimated seroprevalence was >70% in Gauteng on the basis of a population-based survey that was conducted between October and December 2021 (ref. [13]). The weeks following the third wave in South Africa, between 10 October and 15 November 2021, were marked by lower levels of transmission, as indicated by a low incidence of reported COVID-19 cases (100–200 new cases per day) and low (<2%) test positivity rates (Fig. 1a–c).

A rapid increase in COVID-19 cases was observed from the middle of November 2021 in Gauteng province, the economic hub of South Africa containing the cities of Tshwane (Pretoria) and Johannesburg. Specifically, rising case numbers and test positivity rates were first noticed in Tshwane, initially associated with outbreaks in higher-education settings. This resurgence of cases was accompanied by an increasing frequency of S-gene target failure (SGTF) during TaqPath-based diagnostic PCR testing: a phenomenon that was previously observed with the Alpha variant due to a deletion at amino acid positions 69 and 70 (Δ69–70) in the SARS-CoV-2 spike protein[14]. Given the low prevalence of Alpha in South Africa (Fig. 1a), targeted whole-genome sequencing of these specimens was prioritized.

On 19 November 2021, sequencing results from a batch of 8 SGTF samples collected between 14 and 16 November 2021 indicated that all were of a new and genetically distinct lineage of SARS-CoV-2. Further rapid sequencing identified the same variant in 29 out of 32 routine diagnostic samples from multiple locations in Gauteng province, indicating the widespread circulation of this new variant by the second week of November. Crucially, this rise immediately preceded a sharp increase in reported case numbers (Fig. 1c, Extended Data Fig. 1). In the following four days, the presence of this lineage was confirmed by sequencing in another two provinces—KwaZulu-Natal and the Western Cape (Fig. 1b).

Concurrently, in Gaborone, Botswana (<360 km from Tshwane), four genomes generated from samples collected on 11 November 2021 and sequenced on 17–18 November 2021 as part of weekly surveillance displayed an unusual set of mutations. These were reported to the Botswana Ministry of Health and Wellness on 22 November 2021 as unusual sequences that were linked to a group of visitors (non-residents) on a diplomatic mission. The sequences were uploaded to GISAID[15,16] on 23 November 2021, and it became apparent that they belonged to a new lineage. A further 15 genomically confirmed cases (not epidemiologically linked to the first four) were identified within the same week from various other locations in Botswana. All of these either had travel links from South Africa, or were contacts of someone with travel links.

On 24 November 2021, these SARS-CoV-2 genomes from both South Africa and Botswana were designated as belonging to a new PANGO lineage (B.1.1.529)[17], which was later divided into sublineages aliased BA.1 (the main clade), BA.2 and BA.3. On 26 November 2021, the lineage was designated a VOC and named Omicron by the WHO on the recommendation of the Technical Advisory Group on SARS-CoV-2

Virus Evolution[18]. By the first week of December 2021, Omicron was causing a rapid and sustained increase in cases in South Africa and Botswana (Fig. 1c, Extended Data Fig. 2 (for Botswana)). In Gauteng, weekly test positivity rates increased from <1% in the week beginning 31 October, to 16% in the week beginning 21 November 2021, and to 35% in the week beginning 28 November, concurrent with an exponential rise in COVID-19 incidence (Fig. 1c, Extended Data Fig. 1). Nationally, daily case numbers exceeded 22,000 (84% of the peak of the previous wave of infections) by 9 December 2021. At the same time, the proportion of TaqPath PCR tests with SGTF increased rapidly in all provinces of South Africa, reaching ~90% nationally by the week beginning 21 November 2021, strongly indicating that the fourth wave was being driven by Omicron—an indication that has now been confirmed by virus genome sequencing in all provinces (Fig. 1c). Similarly, Botswana experienced a sharp increase in cases, doubling every 2–3 days during late November to early December 2021, transitioning from a 7-day moving average of <10 cases per 100,000 individuals to above 25 cases per 100,000 individuals in less than 10 days (Extended Data Fig. 2).

By 16 December 2021, Omicron had been detected in 87 countries, both in samples from travellers returning from southern Africa, and in samples from routine community testing (Extended Data Fig. 3) and, by 1 January 2022, over 100,000 genomes had been produced from over 100 countries and Omicron was becoming the dominant VOC in the world.

## Evolutionary origins of Omicron

To determine when and where Omicron probably originated, we analysed all 686 available Omicron genomes (including 248 from southern Africa and 438 from elsewhere in the world) retrieved from GISAID (date of access, 7 December 2021)[15,16], in the context of a global reference set of representative SARS-CoV-2 genomes (n = 12,609) collected between December 2019 and November 2021. Preliminary maximum-likelihood phylogenies identified the Omicron BA.1 sequences as a monophyletic clade rooted within the B.1.1 lineage (Nextstrain clade 20B), with no clear basal progenitor (Fig. 2a). Importantly, the BA.1 cluster is highly phylogenetically distinct from any known VOCs or variants of interest (VOIs) and from any other lineages that are known to be circulating in southern Africa (such as C.1.2) (Fig. 2a). More recently, two related lineages have emerged (BA.2 and BA.3), both sharing many, but not all of the characteristic mutations of BA.1 and both having many unique mutations of their own (Extended Data Fig. 4a, b). While BA.2 and BA.3 are evolutionarily linked to BA.1 in that they all branch off of the same B.1.1 node without obvious progenitors, the three sublineages evolved independently from one another along separate branches (Extended Data Fig. 4c, d). The earliest specimens of BA.2 and BA.3 were both sampled after the earliest known BA.1 in South Africa (8 November 2021 at the time of writing), on 17 November 2021 in Tshwane (Gauteng) and on 18 November 2021 in a neighbouring province (North West), respectively. We primarily focus here on the BA.1 lineage, which is rapidly spreading in multiple countries around the world and is the lineage that was first officially designated as the Omicron VOC.

Time-calibrated Bayesian phylogenetic analysis of all BA.1 assigned genomes from southern Africa (as of 11 December 2021, n = 553) estimated the time at which the most recent common ancestor (TMRCA) of the analysed BA.1 lineage sequences existed to be 9 October 2021 (95% highest posterior density (HPD) 30 September–20 October) with a per-day exponential growth rate of 0.137 (95% HPD = 0.099–0.175) reflecting a doubling time of 5.1 days (95% HPD = 4.0–7.0) (Fig. 2b). These estimates are robust to whether the evolutionary rate is estimated from the data or fixed to previously estimated values (Extended Data Table 1). Limiting the analysis to genomes from Gauteng province only yields a faster growth rate estimate with a doubling time of

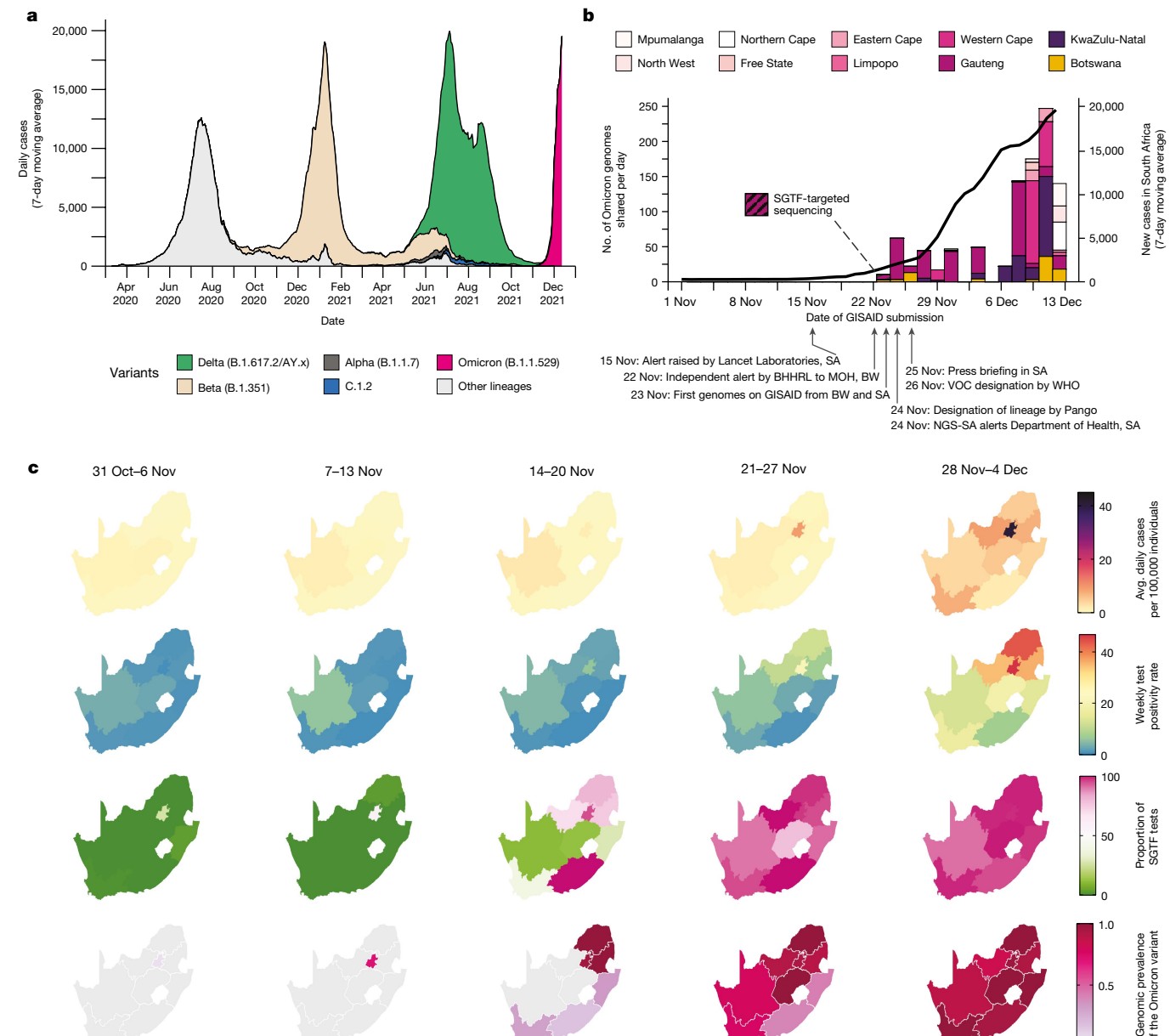

**Fig. 1 | Detection of Omicron variant. a**, The progression of daily reported cases in South Africa from March 2020 to December 2021. The 7-day rolling average of daily case numbers is coloured by the inferred proportion of variants responsible for the infections, as calculated by genomic surveillance data on GISAID. **b**, Timeline of Omicron detection in Botswana and South Africa. Bars represent the number of Omicron genomes shared per day, according to the date they were uploaded to GISAID; the line represents the 7-day moving average of daily new cases in South Africa. BHHRL, Botswana Harvard HIV Reference Laboratory; BW, Botswana; NGS-SA, Network for Genomic Surveillance in South Africa; SA, South Africa. **c**, Weekly progression of average daily cases per 100,000 individuals, test positivity rates, proportion of SGTF tests (on the TaqPath COVID-19 PCR assay) and genomic prevalence of Omicron in nine provinces of South Africa for five weeks from 31 October to 4 December 2021. Note that, because of heterogeneous use of the TaqPath PCR assay across provinces, the proportion of SGTF tests illustrated for the Eastern Cape province in weeks of 14–20 November and 21–27 November 2021 are based on only 2 and 4 data points, respectively. Genomic prevalence here is equivalent to the proportion of weekly surveillance sequences genotyped as being Omicron.

2.8 days (95% HPD = 2.1–4.2) (Extended Data Table 1). Using a phylo-dynamic model that accounts for variable genome sampling through time (birth–death skyline model (BDSKY)[19]) yields a doubling time of BA.1-assigned genomes from South Africa and Botswana ($n$ = 552) of 3.9 (95% HPD = 3.5–4.3) days with an effective reproduction number ($R_e$) of 2.79 (95% HPD = 2.60–2.97) during the period from early November to early December. The BDSKY-estimated $R_e$ for the Gauteng province dataset is 3.86 (95% HPD = 3.43–4.29) and 3.61 (95% HPD = 3.20–4.02) for the 3-epoch and 4-epoch model, respectively

(Extended Data Tables 4 and 5). Spatiotemporal phylogeographic analysis indicates that the BA.1 variant spread from the Gauteng province of South Africa to seven of the eight other provinces and to two regions of Botswana from late October to late November 2021, and shows evidence of more recent transmission within and between other South African provinces (Fig. 2c). However, this does not imply that Omicron originated in Gauteng and these phylogeographic infer-ences could change as further genomic data accumulate from other locations.

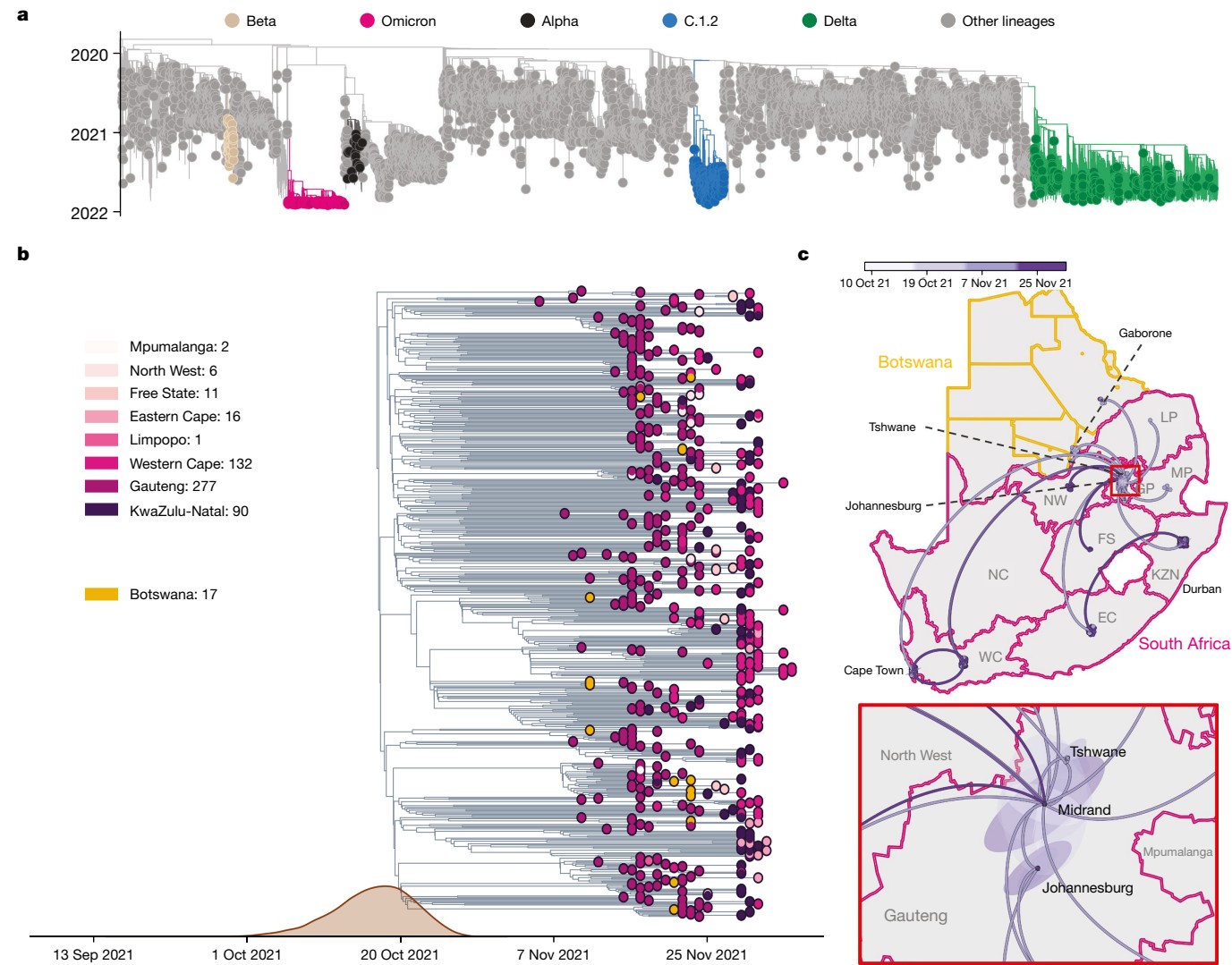

**Fig. 2 | Evolution of Omicron. a**, Time-resolved maximum likelihood phylogeny of 13,295 SARS-CoV-2 genomes; 9,944 of these are from Africa (denoted with tip point circle shapes). Alpha, Beta and Delta VOCs and the C.1.2 lineage, recently circulating in South Africa, are denoted in black, brown, green and blue, respectively. The newly identified SARS-CoV-2 Omicron variant is shown in pink. Genomes of other lineages are shown in grey. **b**, Time-resolved maximum clade credibility phylogeny of the Omicron cluster of southern African genomes (*n* = 553), with locations indicated. The posterior distribution of the TMRCA is also shown. **c**, Spatiotemporal reconstruction of the spread of the Omicron variant in southern Africa with an inset of Gauteng province. Circles represent nodes of the maximum clade credibility phylogeny, coloured according to their inferred time of occurrence (scale in the top panel). Shaded areas represent the 80% HPD interval and depict the uncertainty of the phylogeographical estimates for each node. Solid curved lines denote the links between nodes and the directionality of movement is anticlockwise along the curve. EC, Eastern Cape; FS, Free State; GP, Gauteng; KZN, KwaZulu-Natal; LP, Limpopo; MP, Mpumalanga; NC, Northern Cape; NW, North West; WC, Western Cape.

## Molecular profile of Omicron

Compared with Wuhan-Hu-1, BA.1 carries 15 mutations in the spike receptor-binding domain (RBD) (Fig. 3), five of which (G339D, N440K, S477N, T478K and N501Y) have been shown individually to enhance bind to human ACE2 (hACE2)[20]. Seven of the RBD mutations (K417N, G446S, E484A, Q493R, G496S, Q498R and N501Y) are expected to have moderate to strong effects on the binding of at least three out of the four major classes of RBD-targeted neutralizing antibodies[21–23]. These RBD mutations coupled with four amino acid substitutions (A67V, T95I, G142D and L212I), three deletions (69–70, 143–145 and 211) and an insertion (EPE between 214 and 215) in the N-terminal domain (NTD)[24] are predicted to underlie the substantially reduced sensitivity of Omicron to neutralization by anti-SARS-CoV-2 antibodies induced by either infection or vaccination[25,26]. These mutations also involve key structural epitopes that are targeted by some of the currently authorized monoclonal antibodies, particularly bamlanivimab + etesevimab and casirivimab + imdevimab[26–29]. Preliminary analysis suggests that, although the spike mutations involve a number of T cell and B cell epitopes, the majority of epitopes (>70%) remain unaffected[30].

Omicron also has a cluster of three mutations (H655Y, N679K and P681H) adjacent to the S1/S2 furin cleavage site (FCS) that are likely to enhance spike protein cleavage and fusion with host cells[31,32] and that could also contribute to enhanced transmissibility[33] (Extended Data Fig. 5).

Outside of the spike protein, a deletion in nsp6 (del105–107), in the same region as deletions seen in Alpha, Beta, Gamma and Lambda, may have a role in evasion of innate immunity[34], and the double mutation in nucleocapsid (R203K and G204R)—which is also present in Alpha, Gamma and C.1.2—has been associated with enhanced infectivity in human lung cells[35].

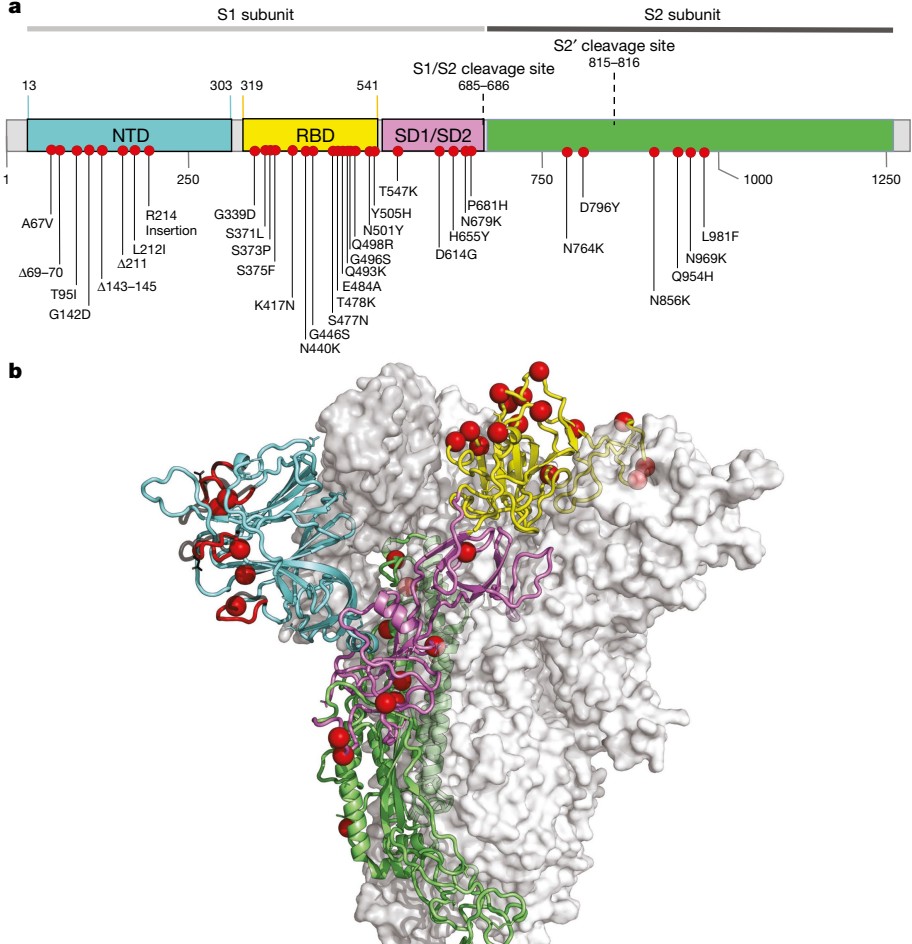

**Fig. 3 | Molecular profile of BA.1. a**, Amino acid mutations on the spike gene of the BA.1 variant. **b**, The structure of the SARS-CoV-2 spike trimer, showing a single spike protomer in cartoon view. The NTD, RBD, subdomains 1 and 2, and the S2 protein are shown in cyan, yellow, pink, and green, respectively. The red spheres indicate the alpha carbon positions for each omicron variant residue. NTD-specific loop insertions/deletions are shown in red, with the original loop shown in transparent black.

## Recombination analysis

Given the large number of mutations differentiating BA.1, BA.2 and BA.3 from other known SARS-CoV-2 lineages, it was considered plausible that (1) all of these lineages might have descended from a common recombinant ancestor; (2) one or more of the BA lineages might have originated through recombination between a virus in one of the other BA lineages and a virus in a non-BA lineage; or (3) one of the BA lineages may have originated through recombination between viruses in the other two BA lineages. We tested these hypotheses using a variety of recombination detection approaches (implemented using GARD[36], 3SEQ[37] and RDP5 (ref. [38])) to identify potential signals of recombination in sequence datasets containing the BA.1, BA.2 and BA.3 sequences together with sequences representative of global SARS-CoV-2 genomic diversity.

Potential evidence of a single recombination event involving BA.1, BA.2 and BA3 was identified by 3SEQ ($P = 0.03$), GARD (delta c-AIC = 20) and RDP5 (GENECONV $P = 0.027$; RDP $P = 0.006$) within the NTD encoding region of spike. The most likely breakpoint locations for this recombination event were 21690 for the 5′ breakpoint (high likelihood interval between 15716 and 21761) and 22198 for the 3′ breakpoint (high likelihood interval between 22197 and 22774). However, these analyses could not reliably identify which of BA.1, BA.2 or BA.3 was the recombinant. Phylogenetic analysis of the genome regions bounded by these breakpoints (genome coordinates 1–21689, 21690–22198 and 22199–29903) potentially supported (1) BA.1 having acquired the NTD encoding region of BA.3 through recombination, (2) BA.3 having acquired the NTD-encoding region of BA.1 through recombination or (3) BA.2 having acquired the NTD-encoding region of a non-BA lineage virus through recombination (Extended Data Fig. 6).

Although we found weak statistical and phylogenetic evidence of one of BA.1, BA.2 or BA.3 being recombinant, we found no evidence that the MRCA of the BA.1, BA.2 and BA.3 lineages was recombinant. However, note that recombination tests in general will not have sufficient statistical power to reliably identify evidence of individual recombination events that result in transfers of less than ~5 contiguous polymorphic nucleotide sites between genomes[36,39,40]. Furthermore, if BA.1, BA.2 and/or BA.3 are the products of a series of multiple partially overlapping recombination events occurring across multiple temporally clustered replication cycles, the complex patterns of nucleotide variation that might result could be extremely difficult to interpret as recombination using the methods applied here[41].

## Selection analysis of Omicron

The large numbers of mutations seen in the BA.1, BA.2 and BA.3 lineage sequences may have accrued at an accelerated pace under the influence of positive selection. To test for evidence of this, we applied a selection analysis pipeline to all of the available sequences designated as BA.1, BA.2 and BA.3 in GISAID as of 20 December 2021. We ran selection

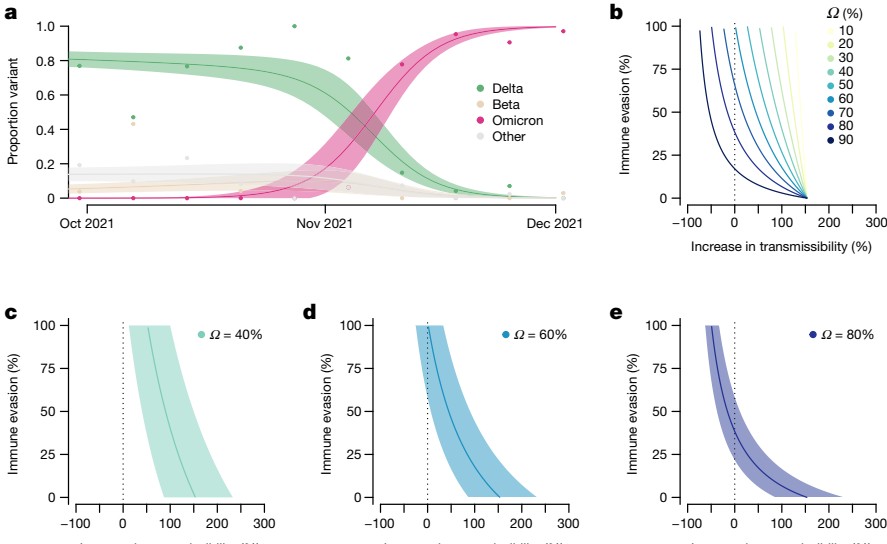

**Fig. 4 | Growth of Omicron in Gauteng, South Africa, and the relationship between potential increase in transmissibility and immune evasion.**
**a**, Omicron rapidly outcompeted Delta in November 2021. Model fits are based on a multinomial logistic regression. Dots represent the weekly proportions of variants. **b**, The relationship between the potential increase in transmissibility and immune evasion strongly depends on the assumed level of current population immunity against Delta that is afforded by previous infections during earlier epidemic waves and/or vaccination ($\Omega$). **c**–**e**, The relationship for a population immunity of 40% (**c**), 60% (**d**) and 80% (**e**) against infection and transmission with Delta. The dark vertical dashed line indicates equal transmissibility of Omicron compared to Delta. The shaded areas correspond to the 95% CIs of the model estimates.

screens individually on BA.1, BA.2 and BA.3 sequences, according to a previously described procedure[34]. We downsampled alignments of individual protein-encoding regions to obtain a median of 110 genetically unique BA.1 sequences, 3 BA.2 sequences, 2.5 BA.3 sequences and around 100 other unique sequences for each gene/open reading frame (ORF) from a representative collection of other SARS-CoV-2 lineages (used as background sequences to contextualize evolution within the Omicron subclade).

Given that the BA.1 lineage has 1,000-fold more sequences than BA.2 and BA.3, we performed the most detailed analysis on it. We detected evidence of gene-wide positive selection (using the BUSTED method[42]) acting on 11 genes or ORFs since the ancestral BA.1 lineage split from the B.1.1 lineage: *M* gene ($P = 0.002$), *N* gene ($P = 0.006$), *nsp3* ($P = 0.05$), *S* gene, exonuclease, *RdRp*, methyltransferase, helicase, *ORF7a*, *ORF6* and *ORF3a* ($P < 0.0001$ for all tests). In all ten genes, this selection was strong (ratio of non-synonymous to synonymous substitutions ($dN/dS$) > 5) and occurred in bursts (≤6% of branch–site combinations selected). The branch separating BA.1 from its most recent B.1.1 ancestor had the most prominent selection signal (which was strongest in the *S* gene, with evidence for nine positively selected sites along this branch[43]), strongly supporting the hypothesis that adaptive evolution had a substantial role in the mutational divergence of Omicron from other B.1.1 SARS-CoV-2 lineages. Relative to the intensity of selection evident within the background B.1.1 lineages, selection in five genes was probably significantly intensified in the BA.1 lineage: *S* gene (intensification factor $K = 2.1$, $P < 0.0001$[44]), exonuclease ($K = 3.50$, $P = 0.0009$), *nsp6* ($K = 2.4$, $P = 0.03$), *RdRp* ($K = 1.14$, $P = 0.02$) and *M* ($K = 4.6$, $P < 0.0001$).

Among 1,546 codon sites that are polymorphic among the BA.1 sequences analysed, 45 were found to have experienced episodic positive selection since BA.1 split from the B.1.1 lineage[45] (MEME $P \le 0.01$; Extended Data Table 2). Twenty-three (51%) of these codon sites are in the *S* gene, thirteen of which contain BA.1-lineage-defining mutations (that is, these selection signals reflect mutations that occurred within the ancestral Omicron lineage). The three positively selected codon sites that did not correspond to sites of lineage-defining mutations (*S*, 346; *S*, 452; and *S*, 701) are particularly notable as these are attributable to mutations that have occurred since the MRCA of the analysed

BA.1 sequences. The mutations driving the positive selection signals at these three sites in the Omicron *S* gene converge on mutations seen in other VOCs or VOIs (R346K in Mu, L452R in Delta, and A701V in Beta and Iota). The A701V mutation, the precise impact of which is currently unknown, is one of 19 in a proposed '501Y-lineage spike meta-signature' comprising the set of mutations that were most adaptive during the evolution of the Alpha, Beta and Gamma VOC lineages[34]. Furthermore, both R346K and L452R are known to affect antibody binding[22] and both of the codon sites at which these mutations occur display evidence of directional selection (using the FADE method[46]). These selective patterns suggest that, during its current rapid spread, BA.1 may be undergoing additional evolution to modify its neutralization profile.

As the numbers of available BA.2 and BA.3 sequences are much lower than for BA.1, the power to perform selection detection was much reduced and not possible for some genomic regions. Nonetheless, there was a strong signal of selection on the *S* gene ($P < 0.0001$ for BA.2 and $P = 0.05$ for BA.3) and selective pressures on this gene in the BA.2 clade were intensified relative to reference SARS-CoV-2 isolates ($K = 6.25$, $P = 0.005$). Within BA.2 sequences, positive selection was detectable on five sites in the *S* gene (371, 376, 405, 477 and 505—all clade defining sites) as well on two sites in the *M* gene (19 and 63—both clade-defining sites). Within BA.3 sequences, positive selection was detectable on four sites in the *S* gene (67, 371, 477 and 505—all clade-defining sites) as well as two sites in the *N* gene (13 and 413—both clade defining sites).

## Transmissibility and immune evasion

We estimated that Omicron had a growth advantage of 0.24 (95% CI = 0.16–0.33) per day over Delta in Gauteng, South Africa (Fig. 4a). This corresponds to a 5.4-fold (95% CI = 3.1–10.1) weekly increase in cases compared with Delta. The growth advantage of Omicron is likely to be mediated by (1) an increase relative to other variants in its intrinsic transmissibility, (2) an increase relative to other variants in its ability to infect, and be transmitted from, previously infected and vaccinated individuals; or (3) both.

The predicted combination of transmissibility and immune evasion for Omicron strongly depends on the assumed level of current

population immunity against infection by, and transmission of, the competing variant Delta that is afforded by previous infections with Beta, Delta and other strains during the three previous epidemic waves in South Africa, and/or vaccination (Fig. 4b). For moderate levels of population immunity against Delta ($\Omega = 0.4$), immune evasion alone cannot explain the observed growth advantage of Omicron (Fig. 4c). For medium levels of immunity against Delta ($\Omega = 0.6$), very high levels of immune evasion could explain the observed growth advantage without an additional increase in transmissibility (Fig. 4d). For high levels of population immunity against Delta ($\Omega = 0.8$), even moderate levels of immune evasion (~25–50%) can explain the observed growth advantage without an additional increase in transmissibility (Fig. 4e). The results of seroprevalence studies and vaccination coverage (~40% of the adult population in South Africa) suggest that the proportion of the population with potential immunity against Delta and earlier variants is probably above 60% (refs. [11,12]). We therefore argue that the population level of protective immunity against Delta acquired during previous epidemic waves is high, and that partial immune evasion is a major driver for the observed dynamics of Omicron in South Africa. This notion is supported by recent findings that show an increased risk of SARS-CoV-2 reinfection associated with the emergence of Omicron in South Africa[47] and the initial results from neutralization assays[48]. However, in addition to immune evasion, an increase or decrease in the transmissibility of Omicron compared with Delta cannot be ruled out.

There are a number of limitations to this analysis. First, we estimated the growth advantage of Omicron based on early sequence data only. These data could be biased due to targeted sequencing of SGTF samples and stochastic effects (such as superspreading) in a low-incidence setting, which can lead to overestimates of the growth advantage and, consequently, of the increased transmissibility and immune evasion. Second, without reliable estimates of the level of protective immunity against Delta in South Africa, we cannot obtain precise estimates of transmissibility or immune evasion of Omicron.

## Conclusion

Strong genomic surveillance systems in South Africa and Botswana enabled the identification of Omicron within a week of observing a resurgence in cases in Gauteng province. Immediate notification of the WHO and early designation as a VOC has stimulated global scientific efforts and has given other countries time to prepare their response. Omicron is now driving a fourth wave of the SARS-CoV-2 epidemic in southern Africa, and is spreading rapidly in several other countries. Genotypic and phenotypic data suggest that Omicron has the capacity for substantial evasion of neutralizing antibody responses, and modelling suggests that immune evasion could be a major driver of the observed transmission dynamics. Close monitoring of the spread of Omicron in countries outside southern Africa will be necessary to better understand its transmissibility and the capacity of this variant to evade post-infection and vaccine-elicited immunity. Neutralizing antibodies are only one component of the immune protection from vaccines and prior infection, and the cellular immune response is predicted to be less affected by the mutations in Omicron. Vaccination therefore remains critical to protect those who have the highest risk of severe disease and death. The emergence and rapid spread of Omicron poses a threat to the world and a particular threat in Africa, where fewer than one in ten people are fully vaccinated.

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

¹Lancet Laboratories, Johannesburg, South Africa. ²Botswana Harvard AIDS Institute Partnership, Botswana Harvard HIV Reference Laboratory, Gaborone, Botswana. ³Harvard T.H. Chan School of Public Health, Boston, MA, USA. ⁴Botswana Presidential COVID-19 Taskforce, Gaborone, Botswana. ⁵National Institute for Communicable Diseases (NICD) of the National Health Laboratory Service (NHLS), Johannesburg, South Africa. ⁶KwaZulu-Natal Research Innovation and Sequencing Platform (KRISP), Nelson R. Mandela School of Medicine, University of KwaZulu-Natal, Durban, South Africa. ⁷South African Medical Research Council Antibody Immunity Research Unit, School of Pathology, Faculty of Health Sciences, University of the Witwatersrand, Johannesburg, South Africa. ⁸Institute of Social and Preventive Medicine, University of Bern, Bern, Switzerland. ⁹Division of Virology, National Health Laboratory Service, Bloemfontein, South Africa. ¹⁰Division of Virology, University of the Free State, Bloemfontein, South Africa. ¹¹Center for Infectious Disease Dynamics, Department of Biology, Pennsylvania State University, University Park, PA, USA. ¹²Diagnofirm Medical Laboratories, Gaborone, Botswana. ¹³Institute of Evolutionary Biology, University of Edinburgh, Edinburgh, UK. ¹⁴Zoonotic Arbo and Respiratory Virus Program, Centre for Viral Zoonoses, Department of Medical Virology, University of Pretoria, Pretoria, South Africa. ¹⁵Emweb, Herent, Belgium. ¹⁶Division of Medical Virology, Faculty of Health Sciences, University of Cape Town, Cape Town, South Africa. ¹⁷Department of Zoology, University of Oxford, Oxford, UK. ¹⁸Department of Biosystems Science and Engineering, ETH Zurich, Zurich, Switzerland. ¹⁹Division of Medical Virology, Faculty of Medicine and Health Sciences, Stellenbosch University, Tygerberg, Cape Town, South Africa. ²⁰Laboratorio de Flavivirus, Fundacao Oswaldo Cruz, Rio de Janeiro, Brazil. ²¹Laboratório de Genética Celular e Molecular, Universidade Federal de Minas Gerais, Belo Horizonte, Brazil. ²²Division of Virology, NHLS Groote Schuur Laboratory, Cape Town, South Africa. ²³Wellcome Centre for Infectious Diseases Research in Africa (CIDRI-Africa), Cape Town, South Africa. ²⁴Division of Computational Biology, Faculty of Health Sciences, University of Cape Town, Cape Town, South Africa. ²⁵Institute for Genomics and Evolutionary Medicine, Department of Biology, Temple University, Philadelphia, PA, USA. ²⁶Health Services Management, Ministry of Health and Wellness, Gaborone, Botswana. ²⁷NHLS Port Elizabeth Laboratory, Port Elizabeth, South Africa. ²⁸Faculty of Health Sciences, Walter Sisulu University, Mthatha, South Africa. ²⁹Public Health Department, Integrated Disease Surveillance and Response, Ministry of Health and Wellness, Gaborone, Botswana. ³⁰Department of Immunology and Infectious Diseases, Harvard T.H. Chan School of Public Health, Boston, MA, USA. ³¹NHLS Tygerberg Laboratory, Tygerberg Hospital, Cape Town, South Africa. ³²Department of Virology, Charlotte Maxeke Johannesburg Academic Hospital, Johannesburg, South Africa. ³³Botswana-Baylor Children's Clinical Centre of Excellence, Gaborone, Botswana. ³⁴Baylor College of Medicine, Houston, TX, USA. ³⁵Department of Medical Virology, University of Pretoria, Pretoria, South Africa. ³⁶National Health Laboratory, Health Services Management, Ministry of Health and Wellness, Gaborone, Botswana. ³⁷National Health Laboratory Service (NHLS), Johannesburg, South Africa. ³⁸Centre for the AIDS Programme of Research in South Africa (CAPRISA), Durban, South Africa. ³⁹Centre for Epidemic Response and Innovation (CERI), School of Data Science and Computational Thinking, Stellenbosch University, Stellenbosch, South Africa. ⁴⁰Department of Medicine, Faculty of Medicine, University of Botswana, Gaborone, Botswana. ⁴¹Department of Medical Laboratory Sciences, School of Allied Health Professions, Faculty of Health Sciences, University of Botswana, Gaborone, Botswana. ⁴²Discipline of Virology, School of Laboratory Medicine and Medical Sciences and National Health Laboratory Service (NHLS), University of KwaZulu-Natal, Durban, South Africa. ⁴³Next Generation Sequencing Unit, Division of Virology, Faculty of Health Sciences, University of the Free State, Bloemfontein, South Africa. ⁴⁴Department of Molecular Medicine and Haematology, University of the Witwatersrand, Johannesburg, South Africa. ⁴⁵School of Pathology, Faculty of Health Sciences, University of the Witwatersrand, Johannesburg, South Africa. ⁴⁶PathCare Vermaak, Pretoria, South Africa. ⁴⁷Institute of Infectious Disease and Molecular Medicine, University of Cape Town, Cape Town, South Africa. ⁴⁸Department of Molecular Pathology, School of Pathology, Faculty of Health Sciences, University of the Witwatersrand, Johannesburg, South Africa. ⁴⁹Department of Global Health, University of Washington, Seattle, WA, USA. ⁵⁰These authors contributed equally: Raquel Viana, Sikhulile Moyo, Daniel G. Amoako, Houriiyah Tegally, Cathrine Scheepers. ⁵¹These authors jointly supervised this work: Simani Gaseitsiwe, Anne von Gottberg, Tulio de Oliveira. ✉e-mail: tulio@sun.ac.za

## Methods

### Epidemiological dynamics

We analysed daily cases of SARS-CoV-2 in South Africa up to 14 December 2021 from publicly released data provided by the National Department of Health and the National Institute for Communicable Diseases. This was accessible through the repository of the Data Science for Social Impact Research Group at the University of Pretoria (https://github.com/dsfsi/covid19za)[49,50]. The National Department of Health releases daily updates on the number of confirmed new cases, deaths and recoveries, with a breakdown by province. Daily case numbers for Botswana were obtained through Our World in Data (OWID) COVID-19 data repository (https://github.com/owid/covid-19-data). We obtained test positivity data from weekly reports from the National Institute for Communicable Diseases (NICD)[51]. Data to calculate the proportion of positive TaqPath COVID-19 PCR tests (Thermo Fisher Scientific) with SGTF in South Africa was obtained from the National Health Laboratory Service and Lancet Laboratories. Test positivity data for Botswana was obtained from the National Health Laboratory up to 6 December 2021. All data visualization was generated through the ggplot package in R[52].

### SARS-CoV-2 sampling

As part of the NGS-SA, seven sequencing hubs in South Africa receive randomly selected samples for sequencing every week according to approved protocols at each site[53]. These samples include remnant nucleic acid extracts or remnant nasopharyngeal and oropharyngeal swab samples from routine diagnostic SARS-CoV-2 PCR testing from public and private laboratories in South Africa. In response to a focal resurgence of COVID-19 in the City of Tshwane Metropolitan Municipality in Gauteng province in November, we enriched our routine sampling with additional samples from the affected area, including initial targeted sequencing of SGTF samples. In Botswana, all public and private laboratories submit randomly selected residual nasopharyngeal and oropharyngeal PCR positive samples weekly to the National Health Laboratory (NHL) and the Botswana Harvard HIV Reference Laboratory (BHHRL) for sequencing.

### Ethical statement

The genomic surveillance in South Africa was approved by the University of KwaZulu-Natal Biomedical Research Ethics Committee (BREC/00001510/2020), the University of the Witwatersrand Human Research Ethics Committee (HREC) (M180832), Stellenbosch University HREC (N20/04/008_COVID-19), University of Cape Town HREC (383/2020), University of Pretoria HREC (H101/17) and the University of the Free State Health Sciences Research Ethics Committee (UFS-HSD2020/1860/2710). The genomic sequencing in Botswana was conducted as part of the national vaccine roll-out plan and was approved by the Health Research and Development Committee (Health Research Ethics body, HRDC#00948 and HRDC#00904). Individual participant consent was not required for the genomic surveillance. This requirement was waived by the Research Ethics Committees.

### Ion Torrent Genexus Integrated Sequencer methodology for rapid whole-genome sequencing of SARS-CoV-2

Viral RNA was extracted using the MagNA Pure 96 DNA and Viral Nucleic Acid kit on the automated MagNA Pure 96 system (Roche Diagnostics) according to the manufacturer's instructions. Extracts were then screened by quantitative PCR to acquire the mean cycle threshold ($C_t$) values for the SARS-CoV-2 *N* and *ORF1ab* genes using the TaqMan 2019-nCoV assay kit v1 (Thermo Fisher Scientific) on the ViiA7 Real-time PCR system (Thermo Fisher Scientific) according to the manufacturer's instructions. Extracts were sorted into batches of $n = 8$ within a $C_t$ range difference of 5 for a maximum of two batches per run. Extracts with <200 copies were sequenced using the low viral titre protocol. Next-generation sequencing was performed using the

Ion AmpliSeq SARS-CoV-2 Research Panel on the Ion Torrent Genexus Integrated Sequencer (Thermo Fisher Scientific), which combines automated cDNA synthesis, library preparation, templating preparation and sequencing within 24 h. The Ion Ampliseq SARS-CoV-2 Research Panel consists of two primer pools targeting 237 amplicons tiled across the SARS-CoV-2 genome providing >99% coverage of the SARS-CoV-2 genome (~30 kb) and an additional five primer pairs targeting human expression controls. The SARS-CoV-2 amplicons range from 125 bp to 275 bp in length. TRINITY was used for de novo assembly and the Iterative Refinement Meta-Assembler (IRMA) was used for genome assisted assembly as well as FastQC for quality checks.

### Whole-genome sequencing and genome assembly

RNA was extracted on an automated Chemagic 360 instrument, using the CMG-1049 kit (Perkin Elmer). The RNA was stored at −80 °C before use. Libraries for whole-genome sequencing were prepared using either the Oxford Nanopore Midnight protocol with Rapid Barcoding or the Illumina COVIDseq Assay.

**Illumina Miseq/NextSeq.** For the Illumina COVIDseq assay, the libraries were prepared according to the manufacturer's protocol. In brief, amplicons were tagmented, followed by indexing using the Nextera UD Indexes Set A. Sequencing libraries were pooled, normalized to 4 nM and denatured with 0.2 N sodium acetate. A 8 pM sample library was spiked with 1% PhiX (PhiX Control v3 adaptor-ligated library used as a control). We sequenced libraries using the 500-cycle v2 MiSeq Reagent Kit on the Illumina MiSeq instrument (Illumina). On the Illumina NextSeq 550 instrument, sequencing was performed using the Illumina COVIDSeq protocol (Illumina), an amplicon-based next-generation sequencing approach. The first-strand synthesis was performed using random hexamers primers from Illumina and the synthesized cDNA underwent two separate multiplex PCR reactions. The pooled PCR amplified products were processed for tagmentation and adapter ligation using IDT for Illumina Nextera UD Indexes. Further enrichment and clean-up was performed according to protocols provided by the manufacturer (Illumina). Pooled samples were quantified using the Qubit 3.0 or 4.0 fluorometer (Invitrogen) and the Qubit dsDNA High Sensitivity assay kit according to the manufacturer's instructions. The fragment sizes were analysed using the TapeStation 4200 (Invitrogen). The pooled libraries were further normalized to 4 nM concentration, and 25 µl of each normalized pool containing unique index adapter sets was combined into a new tube. The final library pool was denatured and neutralized with 0.2 N sodium hydroxide and 200 mM Tris-HCl (pH 7), respectively. Sample library (1.5 pM) was spiked with 2% PhiX. Libraries were loaded onto a 300-cycle NextSeq 500/550 HighOutput Kit v2 and run on the Illumina NextSeq 550 instrument (Illumina).

**Midnight protocol.** For Oxford Nanopore sequencing, the Midnight primer kit was used as described previously[54]. cDNA synthesis was performed on the extracted RNA using the LunaScript RT mastermix (New England BioLabs) followed by gene-specific multiplex PCR using the Midnight primer pools, which produce 1,200 bp amplicons that overlap to cover the 30 kb SARS-CoV-2 genome. Amplicons from each pool were pooled and used neat for barcoding with the Oxford Nanopore Rapid Barcoding kit according to the manufacturer's protocol. Barcoded samples were pooled and bead-purified. After the bead clean-up, the library was loaded on a prepared R9.4.1 flow-cell. A GridION X5 or MinION sequencing run was initiated using MinKNOW software with the base-call setting switched off.

**Genome assembly.** We assembled paired-end and Nanopore .fastq reads using Genome Detective v.1.132 (https://www.genomedetective.com), which was updated for the accurate assembly and variant calling of tiled primer amplicon Illumina or Oxford Nanopore reads, and the Coronavirus Typing Tool[55]. For Illumina assembly, the GATK

HaploTypeCaller --min-pruning 0 argument was added to increase mutation calling sensitivity near sequencing gaps. For Nanopore, low-coverage regions with poor alignment quality (<85% variant homogeneity) near sequencing/amplicon ends were masked to be robust against primer drop-out experienced in the spike gene, and the sensitivity for detecting short inserts using a region-local global alignment of reads was increased. We also used the wf_artic (ARTIC SARS-CoV-2) pipeline as built using the Nextflow workflow framework[56]. In some instances, mutations were confirmed visually with .bam files using Geneious v.2020.1.2 (Biomatters). The reference genome used throughout the assembly process was NC_045512.2 (numbering equivalent to MN908947.3).

Raw reads from the Illumina COVIDSeq protocol were assembled using the Exatype NGS SARS-CoV-2 pipeline v.1.6.1 (https://sars-cov-2.exatype.com/). This pipeline performs quality control on reads and then maps the reads to a reference using Examap. The reference genome used throughout the assembly process was NC_045512.2 (accession number: MN908947.3).

Several of the initial Ion Torrent genomes contained a number of frameshifts, which caused unknown variant calls. Manual inspection revealed that these were probably sequencing errors resulting in mis-assembled regions (probably due to the known error profile of Ion Torrent sequencers)[57]. To resolve this, the raw reads from the Ion Torrent platform were assembled using the SARSCoV2 RECoVERY (Reconstruction of Coronavirus Genomes & Rapid Analysis) pipeline implemented in the Galaxy instance ARIES (https://aries.iss.it). This pipeline fixed the observed frameshifts, confirming that they were artefacts of mis-assembly; this subsequently resolved the variant calls. The Exatype and RECoVERY pipelines each produce a consensus sequence for each sample. These consensus sequences were manually inspected and polished using Aliview v.1.27 (http://ormbunkar.se/aliview/).

All of the sequences were deposited in GISAID (https://www.gisaid.org/)[15,16], and the GISAID accession identifiers are included in Supplementary Table 1. Raw reads for our sequences have also been deposited at the NCBI Sequence Read Archive (BioProject: PRJNA784038).

The number and position of the Omicron mutations has affected a number of primers and caused primer drop-outs across a range of sequencing protocols, especially within the RBD (https://primer-monitor.neb.com/lineages). These primer drop-outs have resulted in a number of genomes missing stretches of the RBD, and can affect estimates of mutation prevalence and the determination of the true set of lineage-defining mutations. Given this, .bam files of all initial genomes were inspected using IG Viewer to confirm mutation calls where reference calls were suspected to be from low coverage at primer dropout sites[58].

**Lineage classification.** We used the widespread dynamic lineage classification method from the Phylogenetic Assignment of Named Global Outbreak Lineages (PANGOLIN) software suite (https://github.com/hCoV-2019/pangolin)[17]. This is aimed at identifying the most epidemiologically important lineages of SARS-CoV-2 at the time of analysis, enabling researchers to monitor the epidemic in a particular geographical region. For the Omicron variant described in this study, the corresponding PANGO lineage designation is BA.1 (lineages v.1.2.106). When first characterized, the lineage was designated B.1.1.529, but the emergence of three sibling lineages to Omicron resulted in the split into sublineages (B.1.1.529.1, B.1.1.529.2 and B.1.1.529.3, aliased as BA.1, BA.2 and BA.3). BA.1 contains all the genomes with the original mutational constellation that was designated as Omicron and, at time of writing, is the dominant sublineage.

**Recombination testing.** To test for the possibility that the Omicron lineage (including BA.1, BA.2 and BA.3) is a recombinant of other SARS-CoV-2 lineages, we used a global subsample of sequences spanning January 2021 to August 2021. Using the NCBI SARS-CoV-2 Data hub[59,60], we constructed a dataset containing 221 sequences by randomly sampling five sequences from each month for each continent. No Oceania samples were available from July or August, and no South American sequences were available from July 2021 (ref. [61]). These sequences were aligned together with a set of five high-quality BA.1, six BA.2 and one BA.3 sequences (representing the known diversity of these clades on 5 December 2021) using MAFFT[62] with the default settings. Whereas 3SEQ[37] and RDP5 (ref. [38]) were used to analyse this dataset, a subsample of the 39 most divergent sequences from the dataset was analysed using the GARD recombination detection method[36]. As none of these recombination detection methods normally use potentially informative deletion patterns, deletions in these alignments were recoded as nucleotide substitutions (one substitution per contiguous run of deleted nucleotides). Furthermore, to minimize multiple testing issues, BA.1, BA.2 and BA.3 were tested for evidence of recombination among one another using individual sequences from each of these lineages (CERI-KRISP-K032254, EPI_ISL_7190366 and EPI_ISL_7526186, respectively) together with the Wuhan-Hu-1 sequence (which served as a reference point for rooting the four taxon phylogeny). The default program settings were used throughout for recombination analyses, with the exception of RDP5 analysis, in which sequences were treated as linear and the window sizes for the SiScan and BootScan methods (two of the seven recombination detection methods applied in RDP5) were changed to 2,000 nucleotides.

**Selection analyses.** We investigated the nature and extent of selective forces acting on BA.1, BA.2 and BA.3 genes encoding individual protein products (respectively, a median of 110, 3 and 2.5 unique BA.1, BA.2 and BA.3 sequences per protein product encoding genome region). A subset of publicly available sequences (from the Virus Pathogen Database and Analysis Resource (ViPR); https://www.viprbrc.org/) was included as background sequences to contextualize selection signals detectable within the BA.1, BA.2 and BA.3 lineages at the levels of complete protein product encoding regions, and individual codons (a median of ~100 sequences per protein coding region). Sequences were selected, quality-checked, aligned, and processed for BUSTED, RELAX, MEME, FADE, FEL and BGM selection analyses (all implemented in HyPhy v.2.5.33)[63] using the automated RASCL pipeline as outlined previously[2,9,34].

**Structure modelling.** We modelled the spike protein on the basis of the Protein Data Bank coordinate set 7A94, showing the first step of the spike protein trimer activation with one RBD domain in the up position, bound to the human ACE2 receptor[64]. We used Pymol (The PyMOL Molecular Graphics System, v.2.2.0) for visualization.

**Phylogenetic analysis.** All sequences on GISAID[15,16] designated Omicron (*n* = 686; date of access: 7 December 2021) were analysed against a globally representative reference set of SARS-CoV-2 genotypes (*n* = 12,609) spanning the entire genetic diversity observed since the start of the pandemic. In brief, the reference set included: (1) all genomes from Africa assigned to PANGO lineage B.1.1 or any of its descendents, excluding those belonging to a VOC clade; (2) a representative subsampling of global data from the publicly maintained global build of Nextstrain (https://nextstrain.org/ncov/gisaid/global); and (3) the top thirty BLAST hits when querying GISAID BLAST for BA.1 and BA.2 sequences. This sampling scheme ensures that we analyse Omicron against the closest variants of the virus. Omicron and reference sequences were aligned using Nextalign[65]. A maximum-likelihood tree topology was inferred in FastTree[66] under the following parameters: a General Time Reversible model of nucleotide substitution and a total of 100 bootstrap replicates[67]. The resulting maximum-likelihood tree topology was transformed into a time-calibrated phylogeny in which branches along the tree were scaled in calendar time using TreeTime[68]. The resulting tree was then visualized and annotated in ggtree in R[69].

Additional BA.2 ($n = 148$) and BA.3 ($n = 19$) sequences were added to the above phylogeny after review to clarify the evolutionary relationship between BA.1, BA.2 and BA.3 (Extended Data Fig. 4c, d).

**Time-calibrated BEAST analysis.** To estimate a time-scale and growth rate from the genome sequencing data, BEAST (v.1.10.4)[70,71] was used to sample phylogenetic trees under an exponential growth coalescent model using a strict molecular clock. All BA.1-assigned genomes from South Africa and Botswana (as of 11 December 2021) were included, with some lower coverage genomes removed, leaving a total of 553 genomes. The single South African BA.2 genome (CERI-KRISP-K032307, EPI_ISL_6795834) was included to help to stabilize the root of the BA.1 clade but the exponential growth coalescent model was applied only to BA.1 (a constant population size coalescent was used for the rest of the tree). The rate of molecular evolution was estimated from the data. Two runs of 100 million iterations were compared to assess convergence, and then post-burnin samples were pooled to summarize parameter estimates.

**Birth–death phylogenetic analysis.** We analysed the full South Africa and Botswana dataset ($n = 552$, all BA.1 assigned), and the reduced dataset containing only Gauteng province genomes ($n = 277$) using the serially sampled birth–death skyline (BDSKY) model[19], implemented in BEAST2 (v.2.5.2)[72]. To allow for changes in genomic sampling intensity shortly after the discovery of the new lineage, we allowed the sampling proportion to vary with time while keeping all other models parameters constant over the study period. The choice of prior distributions for the model parameters is summarized in Extended Data Table 3.

For each analysis, we used an HKY substitution model and a strict clock model with a fixed clock rate of $0.75 \times 10^{-3}$ and $1.1 \times 10^{-3}$ substitutions per site per year (s.s.y.) for the full South Africa and Botswana dataset, and Gauteng province-only dataset, respectively. To allow for comparisons with the exponential growth coalescent model, we also repeated the analyses with clock rates fixed at those estimated from the coalescent analyses ($1.2 \times 10^{-3}$ and $0.3 \times 10^{-3}$ s.s.y.). The mean duration of infectiousness was fixed at 10 days[73,74]. The effective reproduction number, $R_e$, was assumed to be constant through time. The sampling proportion was assumed to be 0 before the collection time of the oldest sample and allowed to change at fixed times that were approximately equidistantly spaced between the oldest sample and the most recent sample. For Markov chain Monte Carlo (MCMC) analyses of the full South Africa and Botswana dataset, the maximum clade credibility tree from the exponential growth coalescent model was used as the starting tree. We kept the tree topology fixed, estimating only internal node heights.

To assess the robustness of our estimates of $R_e$ under different assumptions of temporal variations in the sampling proportion, we repeated the analyses with 3 instead of 4 equidistant change-time points. All of the other model parameters and priors were kept the same.

For each analysis, we ran two independent chains of 100 million MCMC steps and sampled parameters every 10,000 steps. We used Tracer (v.1.7)[75] to evaluate MCMC convergence for each of the individual chains (effective sample size (ESS) > 200), which were then combined using LogCombiner to obtain the final posterior distribution after removing 10% of each chain as burn-in. The results were analysed using the bdskytools package in R (https://github.com/laduplessis/bdskytools).

The resulting estimates for the time of the most recent common ancestor, exponential growth rate and doubling time are summarized in Extended Data Tables 4 and 5. With fixed clock rates of $0.75 \times 10^{-3}$ and $1.1 \times 10^{-3}$ s.s.y. for the full South Africa and Botswana dataset and Gauteng province-only dataset, respectively, the 3-epoch and 4-epoch BDSKY models resulted in similar estimates of the effective reproduction number, $R_e$, for both datasets: 2.74 (95% HPD = 2.56–2.92) and 2.79 (95% HPD = 2.60–2.97) for the South Africa and Botswana dataset, and 3.86 (95% HPD = 3.43–4.29) and 3.61 (95% HPD = 3.20–4.02) for the

Gauteng province-only dataset. Using a faster clock rate led to more recent common ancestors and higher estimates of the effective reproduction number and growth rate.

We examined the sensitivity of our estimates to different assumptions regarding the average duration of infectiousness by repeating the analysis of the South Africa and Botswana dataset with different fixed values of the becoming non-infectious rate: 52.1 per year and 26.1 per year, which translate to an infectious period of 7 and 14 days, respectively. These values were selected as plausible bounds based on the infectious period of asymptomatic cases and the time from symptom onset to two negative RT–PCR tests[74]. The 4-epoch model was used with a fixed clock rate of $0.75 \times 10^{-3}$ s.s.y. in these analyses. For each analysis, we ran three independent chains of 35 million MCMC steps and sampled parameters every 10,000 steps. We used Tracer (v.1.7)[75] to evaluate MCMC convergence for each of the individual chains (ESS > 200), which were then combined using LogCombiner to obtain the final posterior distribution after removing 10% of each chain as burn-in.

The results from the sensitivity analyses showed that our estimates are largely robust to alternative assumptions about the infectious period. On doubling of the mean duration of infectiousness from 7 to 14 days, the TMRCA remained mostly the same (10 October 2021 (95% HPD = 2 October–17 October) compared with 11 October 2021 (95% HPD = 3 October–17 October), while the doubling time shifted from 4.4 (95% HPD = 3.9–5.0) days to 3.5 (95% HPD = 3.2–3.9) days. This change in the doubling time is partially explained by differing estimates of the sampling proportion. For most of the epochs, the sampling proportion increases with the doubling time to explain the same number of sequences observed in each instance, that is, if we assume a shorter average duration of infectiousness, then we infer a slower transmission of which a greater proportion of sequences has been sampled.

**Phylogeographic analysis.** MCMC analyses were run in duplicate in BEAST (v.1.10.4)[70,71] for a total of 100 million iterations sampling every 10,000 steps in the chain. Convergence of runs was assessed in Tracer (v.1.7.1)[75] based on high effective sample sizes (>200) and good mixing in the chains. Maximum clade credibility trees for each run were summarized in TreeAnnotator after discarding the first 10% of the chain as burn in. Finally, the spatiotemporal dispersal of Omicron was mapped using the R package seraphim[76].

**Estimating transmission advantage.** We analysed 805 SARS-CoV-2 sequences from Gauteng, South Africa, that were uploaded to GISAID with sample collection dates from 1 September to 1 December 2021 (ref. [15]). We used a multinomial logistic regression model to estimate the growth advantage of Omicron compared with Delta at the time point at which the proportion of Omicron reached 50% (refs. [77,78]). We fitted the model using the multinom function of the nnet package and estimated the growth advantage using the package emmeans in R.

The difference in the net growth rates (that is, the growth advantage) between a variant (Omicron) and the wild type (Delta) can be expressed as follows:[79]

$$\rho = (1 + \tau)\beta(S + \epsilon(1 - S)) - \beta S,$$

where $\tau$ is the increase of the intrinsic transmissibility, $\epsilon$ is the level of immune evasion, $\beta$ is the transmission rate of the wild type and $S$ is the proportion of the population that is susceptible to the wild type. This relationship can be algebraically solved for $\tau$ and $\epsilon$. We further define $R_w = \beta SD$ as the effective reproduction number of the wild-type with $D$ being the generation time. $\Omega = 1 - S$ corresponds to the proportion of the population with protective immunity against infection and subsequent transmission with the wild type.

We estimated $\epsilon$ for different levels of $\tau$ and $\Omega$. To propagate the uncertainty, we constructed 95% credible intervals (CIs) of the estimates from 10,000 parameter samples of $\rho$, $D$ and $R_w$. We assumed $D$ to be normally

distributed with a mean of 5.2 days and a s.d. of 0.8 days (ref. [80]). We sampled from publicly available estimates of the daily $R_w$ based on confirmed cases during the early growth phase of Omicron in South Africa (1 October–31 October 2021; range = 0.78–0.85) (https://github.com/covid-19-Re)[81].

## Reporting summary

Further information on research design is available in the Nature Research Reporting Summary linked to this paper.

## Data availability

All SARS-CoV-2 whole-genome sequences produced by NGS-SA are deposited in the GISAID sequence database and are publicly available subject to the terms and conditions of the GISAID database. The GISAID accession numbers of sequences used in the phylogenetic analysis, including Omicron and global references, are provided in the Supplementary Table 1. Raw reads for our sequences have also been deposited at the NCBI Sequence Read Archive (SRA) (BioProject: PRJNA784038). Other raw data for this study are provided as a supplementary dataset at our GitHub repository (https://github.com/krisp-kwazulu-natal/SARSCoV2_Omicron_Southern_Africa). The reference SARS-CoV-2 genome (MN908947.3) was downloaded from the NCBI database (https://www.ncbi.nlm.nih.gov/). Other publicly available data used in this study are as follows: NCBI SARS-CoV-2 Data Hub (https://www.ncbi.nlm.nih.gov/sars-cov-2/), Protein Data Bank coordinate set 7A94 (https://www.rcsb.org/), Nexstrain global build (https://nextstrain.org/ncov/gisaid/global), Covid-19 Re repository (https://github.com/covid-19-Re), daily Covid-19 case numbers from the Data Science for Social Impact Research Group at the University of Pretoria (https://github.com/dsfsi/covid19za), daily case numbers from OWID (https://github.com/owid/covid-19-data) and the Virus Pathogen Database and Analysis Resource (ViPR) (https://www.viprbrc.org/).

## Code availability

All input files (such as raw data for figures, alignments or XML files), along with all resulting output files and scripts used in the present study are publicly shared at GitHub (https://github.com/krisp-kwazulu-natal/SARSCoV2_Omicron_Southern_Africa).

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

**Acknowledgements** We thank L. de Gouveia, A. Buys, C. Fourie, N. Duma, M. Ndlovu and other members of the NICD Centre for Respiratory Diseases and Meningitis and Sequencing Core Facility; N. Govender, G. Ntshoe, A. Moipone Shonhiwa, D. Muganhiri, I. Matiea, E. Mathatha, F. Gavhi, T. Mashudu Lamola, M. Makhubele, M. Matjokotja, S. Mdleleni, M. Makhubela from the national SARS-CoV-2 NICD surveillance team for NMCSS case data; F. Mckenna, T. Graham Bell, N. Munava, S. Kwenda, M. Raza Bano and J. Khosa from NICD IT for NMCSS case and test data (in particular, SGTF data); and the following people from the diagnostic laboratories for their assistance: K. Reddy, L. Gounder and C. Naicker from NHLS Inkosi Albert Luthuli Central Hospital Laboratory, S. Korsman from the NHLS Groote Schuur Laboratory, and A. Enoch at NHLS Green Point Laboratory; the staff at the global laboratories who generated and made public the SARS-CoV-2 sequences (through GISAID) used as reference dataset in this study (a complete list of individual contributors of sequences is provided in Supplementary Table 1). The research reported in this publication was supported by the Strategic Health Innovation Partnerships Unit of the South African Medical Research Council, with funds received from the South African Department of Science and Innovation. Sequencing activities at KRISP and CERI were supported in part by the WHO, the National Institutes of Health (NIH) (U01 AI151698) for the United World Antivirus Research Network (UWARN), and the Rockefeller Foundation (grants 2021 HTH 017 and 2020 HTH 062). C.L.A. received funding from the European Union's Horizon 2020 research and innovation programme, project EpiPose (no. 101003688). D.P.M. was funded by the Wellcome Trust (222574/Z/21/Z). R.C. and A.R. acknowledge support from the Wellcome Trust (Collaborators Award 206298/Z/17/Z, ARTIC network) and A.R. from the European Research Council (no. 725422, ReservoirDOCS). V.H. was supported by the Biotechnology and Biological Sciences Research Council (BBSRC) (grant no. BB/M010996/1). A.E.Z., J.T., M.U.G.K. and O.G.P. acknowledge support from the Oxford Martin School. M.U.G.K. acknowledges support from the Rockefeller Foundation, Google.org, and the European Horizon 2020 programme MOOD (no. 874850). M.V. and the members of the Zoonotic Arbo and Respiratory Virus Program, UP was funded through the ANDEMIA G7 Global Health Concept: contributions to improvement of International Health, COVID-19 funds through the Robert Koch Institute. The genomic sequencing at UCT/NHLS is funded from the South African Medical Research Council and Department of Science and Innovation; and by the Wellcome Centre for Infectious Diseases Research in Africa (CIDRI-Africa), which is supported by core funding from the Wellcome Trust (203135/Z/16/Z and 222754). C.W. and J.B. are funded by the EDCTP (RADIATES Consortium; RIA2020EF-3030). Sequencing activities at the NICD were supported by a conditional grant from the South African National Department of Health as part of the emergency COVID-19 response; a cooperative agreement between the National Institute for Communicable Diseases of the National Health Laboratory Service and the United States Centers for Disease Control and Prevention (no. 5U01IP001048-05-00); the African Society of Laboratory Medicine (ASLM) and Africa Centers for Disease Control and Prevention through a

subaward from the Bill and Melinda Gates Foundation grant no. INV-018978; the UK Foreign, Commonwealth and Development Office and Wellcome (no. 221003/Z/20/Z); the South African Medical Research Council (SHIPNCD 76756); the UK Department of Health and Social Care, managed by the Fleming Fund and performed under the auspices of the SEQAFRICA project. The genomic sequencing in Botswana was supported by the Foundation for Innovative New Diagnostics and Fogarty International Center (5D43TW009610), NIH (5K24AI131924-04; 5K24AI131928-05) and support from the Botswana government through the Ministry of Health & Wellness and Presidential COVID-19 Task Force. S. Moyo. was supported in part by the Bill & Melinda Gates Foundation (036530). Under the grant conditions of the Foundation, a Creative Commons Attribution 4.0 Generic License has already been assigned to the Author Accepted Manuscript version that might arise from this submission.

**Author contributions** Genomic data generation: R.V., S. Moyo, D.G.A., H.T., C.S., J.G., J.E., S.G., W.T.C., D.M., B.Z., B.R., L.K., R.S., S.L., M.B.M., P.S.-L., M. Matshaba, M. Mosepele, K. Masupu, A. Mnguni, A. Ismail, B.M., M.S.M., J.E.S., N.N., G. Motsatsi., S.P., G. Marais, T. Mohale, U.R., Y.N., C.W., S.E., T. Maponga, W.P., L. Singh, U.J.A., M. Moir, S.v.W., D.T., K.D., D.H., D.D., R.J., A. Iranzadeh, D.G., P.A.B, M.N., P.N.M. and J.B. Sample collection and metadata curation: R.V., S. Moyo, D.G.A., A. Mendes, A.S., M.D., S. Mayaphi, W.T.C., D.M., P.S.-L., M.C., C.J., L.K.-L., O.L.-A.,

K. Mahlakwane, N.T., N.-Y.H., N. Msomi, K. Moruisi, A.S., A. Maharaj, M.D., Z.M., O.L.-M., Y.R., K.S., D.G., P.A.B., F.T. and M.V. Data analysis: H.T., C.S., R.J.L., N.W., J.E., A.R., C.L.A., E.W., C.K.W., D.P.M., V.H., R.C., J.E.S., M.G., S.P., A.G.L., S.W., M.F.B., A.E.Z., J.T., L.d.P., M.U.G.K. and O.G.P. Study design and data interpretation: R.V., S. Moyo, D.G.A., R.J.L., A.R., C.L.A., S.G., M. Matshaba, M. Mosepele, K. Mlisana, L.K.-L., O.L.-M., M.S.M., K. Moruisi, C.W., L.d.P, O.G.P., A.G., F.T., M.V., J.B., A.v.G. and T.d.O. Manuscript writing: S. Moyo, H.T., R.J.L., J.G., J.E., A.R., C.L.A., E.W., D.P.M., J.B., A.v.G. and T.d.O. All of the authors reviewed the manuscript.

**Competing interests** The authors declare no competing interests.

**Additional information**
**Correspondence and requests for materials** should be addressed to Tulio de Oliveira.

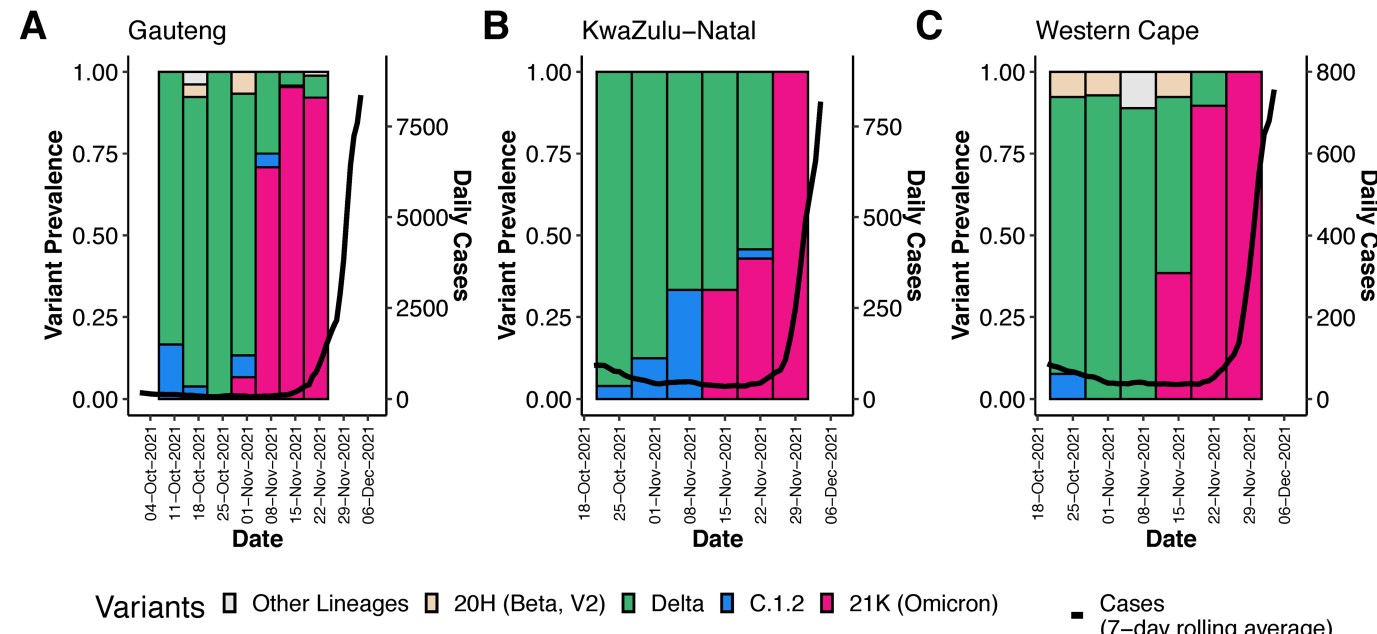

**Extended Data Fig. 1 | Progression of daily recorded cases and variant proportions in Gauteng (A), KwaZulu-Natal (B) and Western Cape (C) provinces between October and December 2021.** A sharp increase in the 7-day rolling average of the number of cases is observed in all three of the biggest provinces in South Africa at the emergence of the Omicron variant.

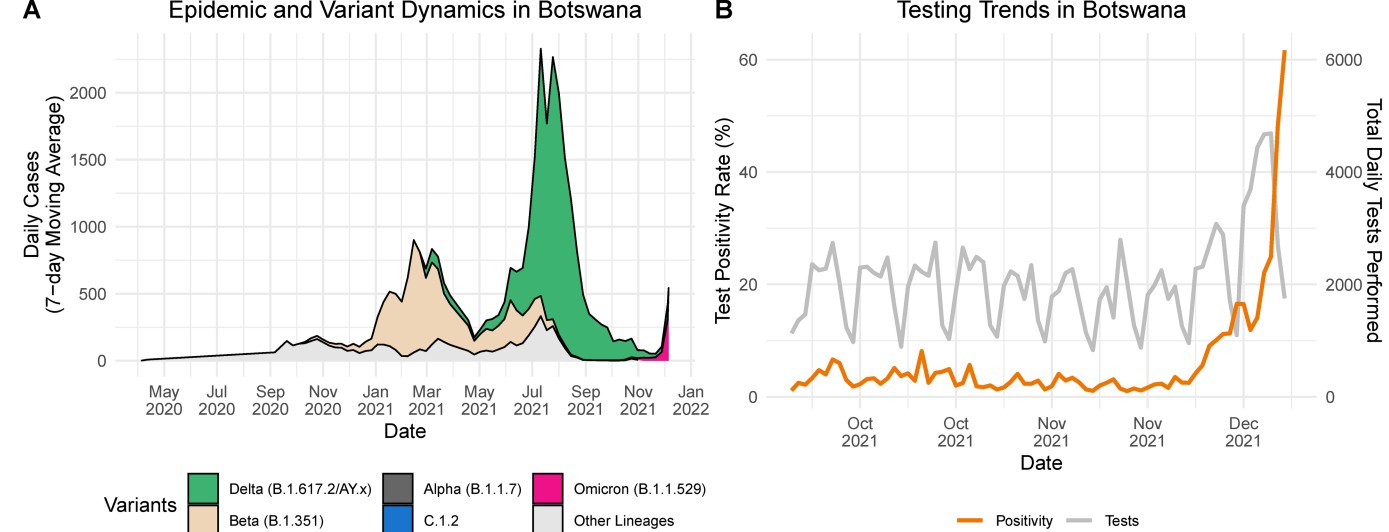

**Extended Data Fig. 2 | Epidemic Progression in Botswana. A**) Epidemic and variant dynamics in Botswana from May 2020 to December 2021, with the 7-day rolling average of the number of recorded cases coloured by the proportion of variants as inferred by genomic surveillance data available on GISAID. At the end of November 2021, a big Delta-driven wave was coming to its end and an Omicron wave was starting at the end of November 2021. **B**) Trends in testing numbers and positivity rates in Botswana between October and December 2021, showing a sharp increase in positivity rate mid-November 2021.

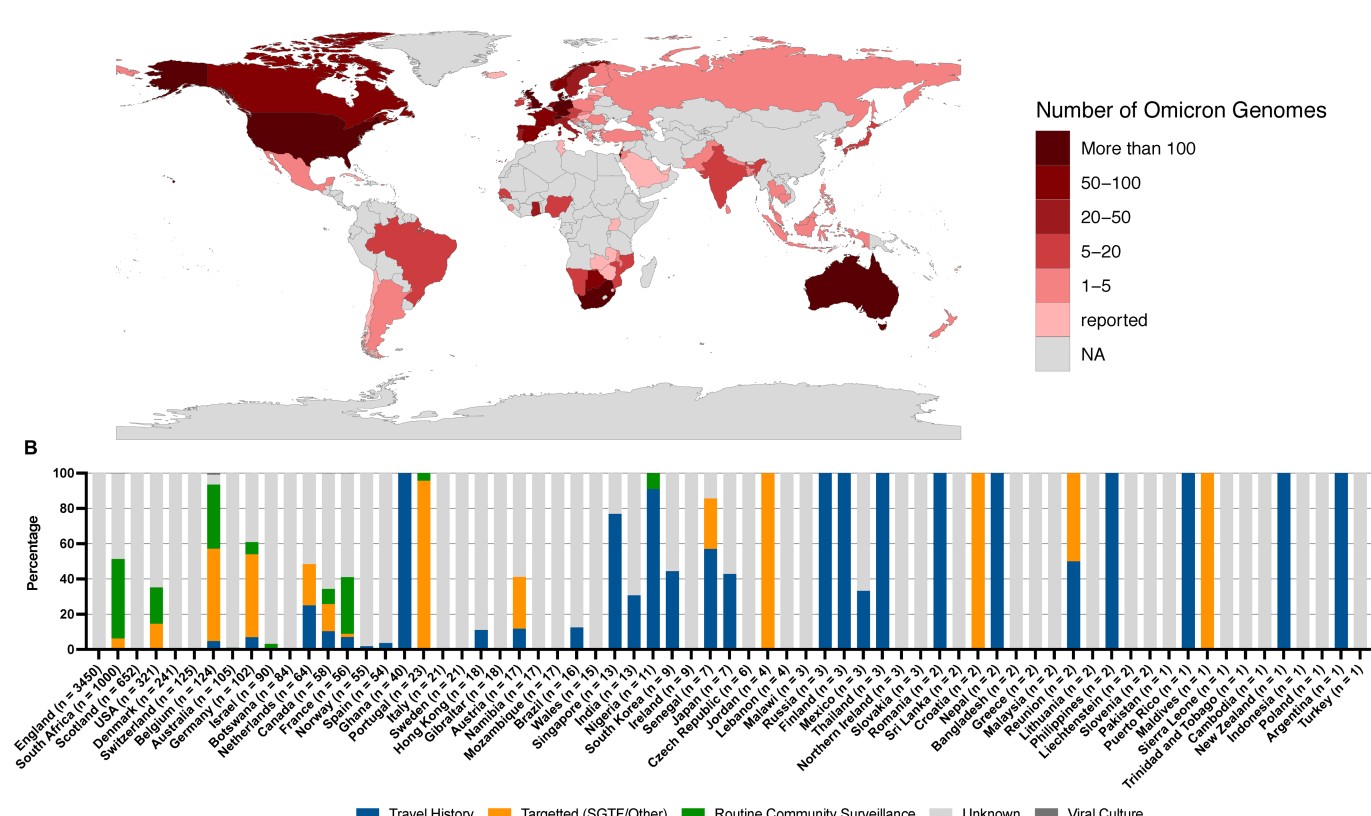

**A** Detection of Omicron Globally (countries = 87; n = 6940)

**Extended Data Fig. 3 | Global distribution of Omicron. (A)** Detection of Omicron globally. Shown are the locations for which Omicron genomes have been deposited on GISAID as of December 16, 2021. Those labelled as "reported" referred to the country from which Omicron has been reported to the WHO but there is currently no sequencing data available in GISAID, all data comes from GISAID and the WHO weekly epidemiology report Edition 70 dated December 14, 2021 (https://reliefweb.int/sites/reliefweb.int/files/resources/20211207_Weekly_Epi_Update_69-%281%29.pdf). Countries are coloured according to the number of genomes deposited with warmer colours representing more genomes. **(B)** Omicron transmission globally. Shown are countries for which Omicron sequencing data is available on GISAID. Proportions of sequences are coloured according to sampling strategy or additional host/location information from either travel history, targeted sequencing (specifically for SGTF, vaccine breakthroughs, outbreaks, contact tracing or other reasons), routine surveillance or unknown if no information has been provided. Countries are ordered by the number of sequences available on GISAID as of December 16, 2021.

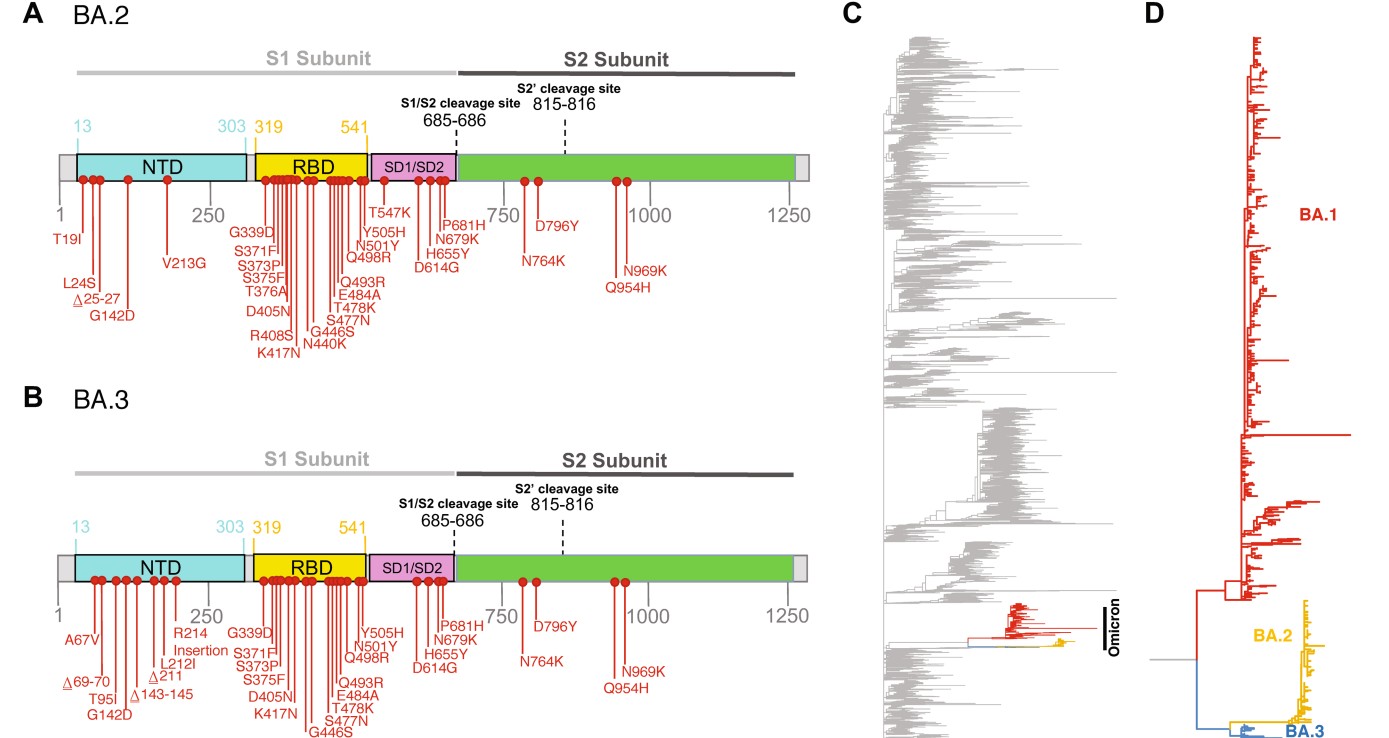

**Extended Data Fig. 4 | Related Lineages BA.2 and BA.3 Molecular Profile and Evolutionary Origins. A**) Amino-acid mutations on the spike gene of the BA.2 **B**) Amino-acid mutations on the spike gene of the BA.3 **C**) Raw maximum likelihood phylogeny of 13,462 SARS-CoV-2 genomes, including 148 BA.2 and 19 BA.3. The newly identified SARS-CoV-2 Omicron variant is shown in colour versus grey for all other lineages. **D**) A zoomed-in view of the Omicron clade showing the evolutionary relationship between BA.1, BA.2 and BA.3.

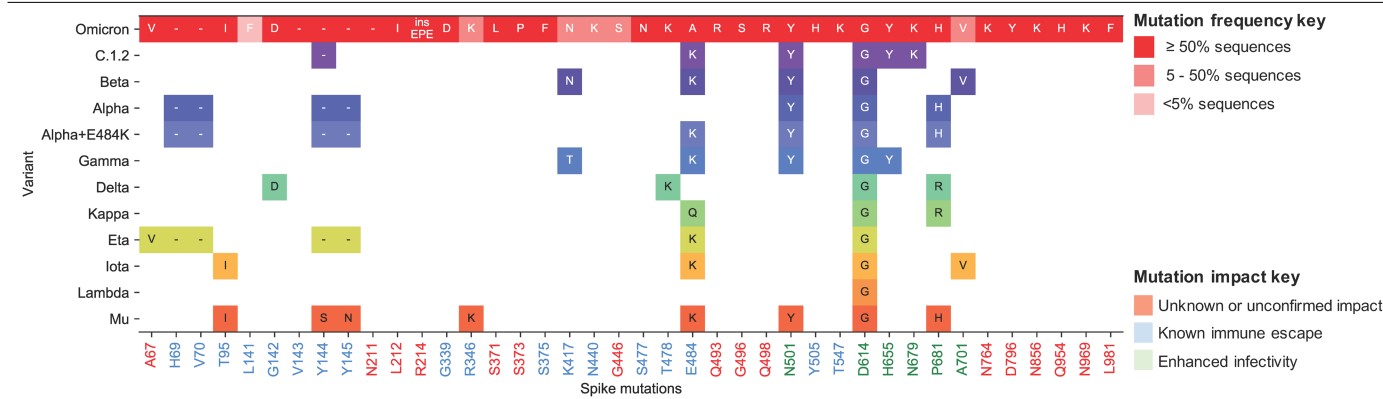

**Extended Data Fig. 5 | BA.1 spike mutations shared with other VOC/VOIs.**
All spike mutations seen in BA.1 are listed at the top in red and coloured according to prevalence. Prevalence was calculated by number of mutation detections / total number of sequences. However, primer drop-outs have affected the RBD region spanning K417N, N440K and G446S, and so it is likely that these mutations may actually be more prevalent than indicated here.

For the VOC/VOIs only mutations that are shared with Omicron and seen in ≥50% of the respective VOC/VOI sequences are shown and are coloured according to Nextstrain clade. The mutations listed at the bottom are shaded according to known immune escape (blue), enhanced infectivity (green) or for unknown/unconfirmed impact (red).

## Region 1: positions 1 - 21690

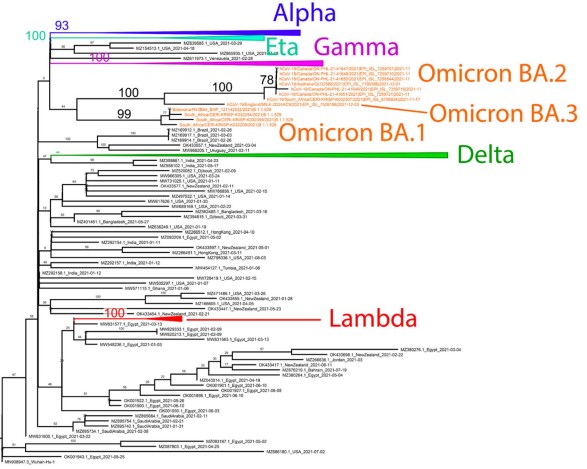

## Region 3: positions 22588 - 30012

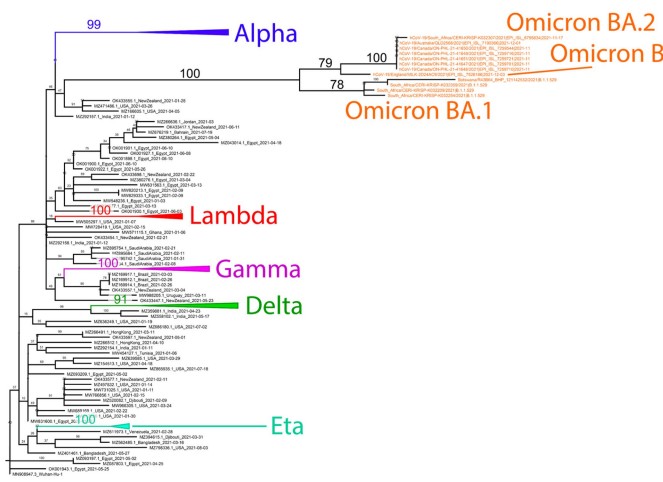

## Region 2: positions 21691 - 22587

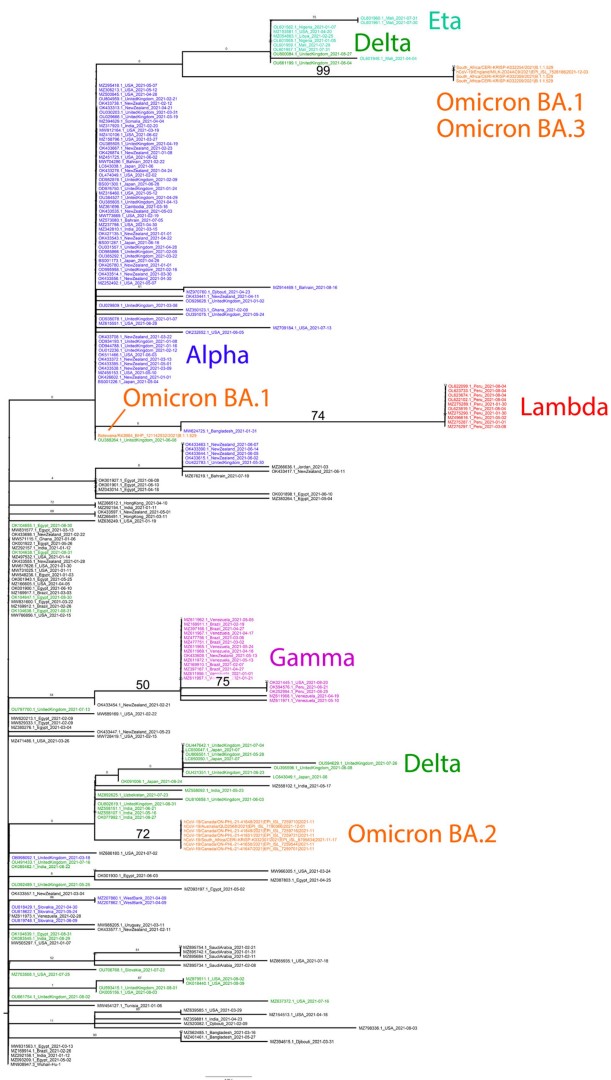

**Extended Data Fig. 6 | Maximum-likelihood trees (inferred with RAxML v8.2.12[82]) for genome regions bounding the consensus recombination breakpoints detected in lineages BA.1, BA.2 and BA.3[83].** The trees include SARS-CoV-2 genome sequences sampled in 2021 (N = 221) together with 13 sequences representing the BA.1, BA.2 and BA.3 lineages. Whereas in trees for regions 1 and 3 BA.2 and BA.3 cluster together with high bootstrap support, BA.1 is a well-supported albeit more distantly related sibling lineage. The a 897nt region 2 segment (encoding the N-terminal domain of spike) includes 67 polymorphic sites with a maximum 8nt difference between strains, showing

little bootstrap support for any sibling or clade relationships except the membership of certain viruses in WHO-designated clades (Lambda, Omicron, Gamma). Despite Omicron lineages BA.1 and BA.3 clustering with certain Delta and Eta viruses and Omicron BA.2 clustering with a distinct set of Delta viruses (all on the basis of several key nucleotide positions), trees based on region 2 show no statistical support for the three Omicron lineages having distinct evolutionary origins. Bootstrap values are shown on branches with relevant values magnified for readability. All trees were rooted on the Wuhan-Hu-1 sequence.

**Extended Data Table 1 | Parameter estimates from BEAST for the full South Africa and Botswana dataset and the reduced data set of only Gauteng Province genomes**

| Data set | Evolutionary rate x10⁻³ changes/site/year | BA.1 Time of most recent common ancestor (TMRCA) | Exponential growth rate (per day) | Doubling time (days) |
|---|---|---|---|---|
| South Africa + Botswana 553 Genomes | 1.20 (0.92, 1.49) | 9 Oct 2021 (30 Sep, 20 Oct) | 0.137 (0.099, 0.175) | 5.1 (4.0, 7.0) |
| South Africa + Botswana 553 Genomes | 1.1 fixed | 8 Oct 2021 (30 Sep, 18 Oct) | 0.137 (0.100, 0.173) | 5.0 (4.0, 7.0) |
| South Africa + Botswana 553 Genomes | 0.75 fixed | 1 Oct 2021 (21 Sep, 13 Oct) | 0.139 (0.099, 0.183) | 5.0 (3.8, 7.0) |
| Gauteng Province, South Africa only 626 genomes 2021-11-05, 2021-12-07 | 0.41 (0.28, 0.54) | 01 Oct 2021 (17 Sept, 17 Oct) | 2.85 (2.10, 4.23) | 2.8 (2.1, 4.2) |
| Gauteng Province, South Africa only 626 genomes 2021-11-05, 2021-12-07 | 1.1 fixed | 19 Oct 2021 (15 Oct, 26 Oct) | 0.29 (0.22, 0.35) | 2.42 (1.96, 3.12) |

95% HPD intervals in parentheses.

**Extended Data Table 2 | Sites in the BA.1 sequences that have been subject to episodic diversifying selection**

| Coordinate (SARS-CoV-2) | Gene/ORF | Codon (in gene/ORF) | # of selected branches | AA composition | p-value | Notes |
|---|---|---|---|---|---|---|
| 3682 | ORF1a | 1140 | 1 | Q/92, L/2 | 0.0061 | |
| 13423 | ORF1a | 4387 | 2 | R/34, H/1, N/1 | 0.0020 | |
| 13627 | ORF1b | 54 | 1 | D/256, -/2, Y/1 | 0.0098 | |
| 18027 | ORF1b | 1520 | 1 | A/171, -/12, Y/1, V/1 | 0.0006 | |
| 18030 | ORF1b | 1521 | 2 | T/171, -/12, K/1, I/1 | 0.0052 | |
| 18267 | ORF1b | 1600 | 1 | E/184, T/1, -/1 | 0.0001 | |
| 18273 | ORF1b | 1602 | 1 | A/184, C/1, -/1 | 0.0001 | |
| 21534 | ORF1b | 2689 | 1 | D/85, S/3 | 0.0066 | |
| 22027 | S | 156 | 3 | E/172, -/11, G/5, P/1 | 0.0006 | |
| 22033 | S | 158 | 1 | R/165, -/23, S/1 | 0.0007 | |
| 22048 | S | 163 | 1 | A/168, -/20, L/1 | 0.0036 | |
| 22072 | S | 171 | 2 | V/167, -/21, K/1 | 0.0000 | |
| 22084 | S | 175 | 1 | F/161, -/26, Q/2 | 0.0000 | |
| 22576 | S | 339 | 3 | D/170, -/11, G/8 | 0.0027 | Clade defining |
| 22597 | S | 346 | 5 | R/151, K/32, -/6 | 0.0007 | Affect Ab binding |
| 22672 | S | 371 | 1 | L/154, S/18, -/16, F/1 | 0.0002 | Clade defining |
| 22678 | S | 373 | 4 | P/149, S/26, -/14 | 0.0009 | Clade defining |
| 22684 | S | 375 | 5 | F/142, S/34, -/13 | 0.0001 | Clade defining |
| 22810 | S | 417 | 5 | N/113, K/41, -/35 | 0.0002 | Clade defining |
| 22879 | S | 440 | 4 | K/120, -/36, N/33 | 0.0018 | Clade defining |
| 22897 | S | 446 | 5 | S/124, -/38, G/27 | 0.0002 | Clade defining |
| 22915 | S | 452 | 4 | L/138, -/36, R/15 | 0.0000 | Affect Ab binding |
| 22990 | S | 477 | 3 | N/148, -/23, S/18 | 0.0005 | Clade defining |
| 23011 | S | 484 | 3 | A/141, -/26, E/21, V/1 | 0.0016 | Clade defining |
| 23047 | S | 496 | 3 | S/151, G/21, -/17 | 0.0051 | Clade defining |
| 23053 | S | 498 | 2 | R/148, -/21, Q/20 | 0.0028 | Clade defining |
| 23074 | S | 505 | 4 | H/142, Y/25, -/22 | 0.0002 | Clade defining |
| 23095 | S | 512 | 1 | V/170, -/18, T/1 | 0.0008 | |
| 23662 | S | 701 | 3 | A/156, V/25, -/7, S/1 | 0.0034 | 501Y metasignature |
| 23851 | S | 764 | 0 | K/150, N/23, -/15, H/1 | 0.0010 | Clade defining |
| 24502 | S | 981 | 3 | F/180, L/6, -/3 | 0.0084 | Clade defining |
| 25548 | ORF3a | 53 | 1 | L/178, F/2 | 0.0099 | |
| 25707 | ORF3a | 106 | 1 | L/158, F/22 | 0.0072 | |
| 26528 | M | 3 | 2 | G/113, -/26, D/9, Y/1 | 0.0041 | |
| 26708 | M | 63 | 3 | T/110, -/28, A/11 | 0.0016 | Clade defining |
| 26765 | M | 82 | 3 | I/111, -/28, T/10 | 0.0019 | |
| 27140 | M | 207 | 1 | N/105, -/42, R/1, S/1 | 0.0011 | |
| 27143 | M | 208 | 2 | T/104, -/42, S/2, I/1 | 0.0066 | |
| 27146 | M | 209 | 1 | D/104, -/42, A/2, Y/1 | 0.0008 | |
| 28253 | ORF8 | 121 | 3 | F/271, I/162, -/24, L/10, V/6, K/5, S/4, Q/1, D/1, C/1 | 0.0013 | |
| 28459 | N | 63 | 2 | D/272, G/11, -/11, Y/1 | 0.0010 | |
| 28471 | N | 67 | 2 | P/280, -/11, S/3, L/1 | 0.0070 | |
| 28477 | N | 69 | 1 | G/282, -/11, K/2 | 0.0001 | |
| 28879 | N | 203 | 3 | K/283, M/8, I/2, -/1, R/1 | 0.0088 | Clade defining |
| 29299 | N | 343 | 3 | D/253, G/40, C/1, H/1 | 0.0002 | |

**Extended Data Table 3 | Prior distributions used for the BDSKY analyses**

| Parameter | Prior distribution | |
|---|---|---|
| | South Africa and Botswana (n = 552) | Gauteng Province only (n = 277) |
| clock rate (x10$^{-3}$ substitutions/site/year) | 0.75 fixed; 1.2 fixed | 1.1 fixed; 0.3 fixed |
| kappa | Lognormal(lnMean = 1, lnSd = 1.25) | |
| gamma shape | Exponential(m = 1) | |
| effective reproduction number | Lognormal(lnMean = 0.8, lnSd = 0.5) | |
| becoming non-infectious rate (per year) | 36.5 fixed | |
| sampling proportion | Beta(alpha = 2, beta = 1000) | Beta(alpha = 2, beta = 100) |
| time of origin | Lognormal(lnMean = -2, lnSd = 0.2) | |

The becoming non-infectious rate was fixed to 36.5/year which corresponds to a mean infectious period of 10 days. A less informative prior for the sampling proportion was used for the Gauteng Province only dataset to allow for the possibility of a higher province-specific sampling proportion.

**Extended Data Table 4 | Time of most recent common ancestor, exponential growth rate and doubling time estimates for the full South Africa and Botswana dataset and the reduced dataset of only Gauteng Province genomes under the 3-epoch BDSKY model in which the sampling proportion was allowed to change at 3 equidistantly spaced time points**

| | Fixed clock rate (x10$^{-3}$ substitutions/site/year) | Time of most recent common ancestor (TMRCA) | Exponential growth rate (per day) | Doubling time (days) |
|---|---|---|---|---|
| South Africa and Botswana (n = 522) | 1.20 | 20 Oct 2021 (13 Oct, 26 Oct) | 0.206 (0.188, 0.226) | 3.4 (3.0, 3.7) |
| | 0.75 | 11 Oct 2021 (3 Oct, 18 Oct) | 0.174 (0.156, 0.192) | 4.0 (3.6, 4.4) |
| Gauteng Province only (n = 277) | 0.30 | 4 Oct 2021 (24 Sep, 12 Oct) | 0.191 (0.151, 0.231) | 3.6 (2.9, 4.5) |
| | 1.1 | 24 Oct 2021 (19 Oct, 29 Oct) | 0.286 (0.243, 0.329) | 2.4 (2.1, 2.8) |

95% HPD intervals in parentheses.

**Extended Data Table 5 | Time of most recent common ancestor, exponential growth rate and doubling time estimates for the full South Africa and Botswana dataset and the reduced dataset of only Gauteng Province genomes under the 4-epoch BDSKY model in which the sampling proportion was allowed to change at 4 equidistantly spaced time points**

| | Fixed clock rate ($\times 10^{-3}$ substitutions/site/year) | Time of most recent common ancestor (TMRCA) | Exponential growth rate (per day) | Doubling time (days) |
|---|---|---|---|---|
| South Africa and Botswana (n = 522) | 1.20 | 19 Oct 2021 (13 Oct, 25 Oct) | 0.205 (0.186, 0.225) | 3.4 (3.1, 3.7) |
| | 0.75 | 11 Oct 2021 (2 Oct, 17 Oct) | 0.179 (0.160, 0.197) | 3.9 (3.5, 4.3) |
| Gauteng Province only (n = 277) | 0.30 | 27 Sep 2021 (16 Sep, 7 Oct) | 0.146 (0.114, 0.180) | 4.8 (3.8, 5.9) |
| | 1.1 | 23 Oct 2021 (17 Oct, 28 Oct) | 0.261 (0.220, 0.302) | 2.7 (2.3, 3.1) |

95% HPD intervals in parentheses.

# nature research

# Reporting Summary

Nature Research wishes to improve the reproducibility of the work that we publish. This form provides structure for consistency and transparency in reporting. For further information on Nature Research policies, see Authors & Referees and the Editorial Policy Checklist.

## Statistics

For all statistical analyses, confirm that the following items are present in the figure legend, table legend, main text, or Methods section.

| n/a | Confirmed | |
|---|---|---|
| ☐ | ☒ | The exact sample size (*n*) for each experimental group/condition, given as a discrete number and unit of measurement |
| ☐ | ☒ | A statement on whether measurements were taken from distinct samples or whether the same sample was measured repeatedly |
| ☐ | ☒ | The statistical test(s) used AND whether they are one- or two-sided *Only common tests should be described solely by name; describe more complex techniques in the Methods section.* |
| ☒ | ☐ | A description of all covariates tested |
| ☐ | ☒ | A description of any assumptions or corrections, such as tests of normality and adjustment for multiple comparisons |
| ☐ | ☒ | A full description of the statistical parameters including central tendency (e.g. means) or other basic estimates (e.g. regression coefficient) AND variation (e.g. standard deviation) or associated estimates of uncertainty (e.g. confidence intervals) |
| ☐ | ☒ | For null hypothesis testing, the test statistic (e.g. *F*, *t*, *r*) with confidence intervals, effect sizes, degrees of freedom and *P* value noted *Give P values as exact values whenever suitable.* |
| ☐ | ☒ | For Bayesian analysis, information on the choice of priors and Markov chain Monte Carlo settings |
| ☐ | ☒ | For hierarchical and complex designs, identification of the appropriate level for tests and full reporting of outcomes |
| ☒ | ☐ | Estimates of effect sizes (e.g. Cohen's *d*, Pearson's *r*), indicating how they were calculated |

*Our web collection on statistics for biologists contains articles on many of the points above.*

## Software and code

Policy information about availability of computer code

| Data collection | No software was used |
|---|---|
| Data analysis | Base-calling for GridIon sequencing was performed on MinKNOW software v21.6. Genome assembly was performed with Genome Detective online tool version 1.132 or Exatype NGS SARS-CoV-2 pipeline v1.6.1 or SARSCoV2 RECoVERY (REconstruction of COronaVirus gEnomes & Rapid analYsis) pipeline implemented in the Galaxy instance ARIES (https://aries.iss.it) and validated with Geneious software v.2020.1.2, IG Viewer or Aliview v1.27. Phylogenetic analysis was performed using FastTree2.1, MAFFT v7.490, Nextalign, BEASTv.1.10.4, BEAST2 v2.5.2, and Tracer v.1.7.1. Selection analyses were performed using HyPhy v2.5.33 through the RASCL pipeline. Recombination analyses were performed using 3SEQ, RDP5 and GARD. Lineage classification was performed using the PANGO software suite (lineages v1.2.106). Structure modeling visualization was performed using PyMOL Molecular Graphics System, version 2.2.0. R packages used for data analysis included ggplot, ggtree, seraphim. Custom codes are all available at: https://github.com/krisp-kwazulu-natal/ SARSCoV2_Omicron_Southern_Africa. |

For manuscripts utilizing custom algorithms or software that are central to the research but not yet described in published literature, software must be made available to editors/reviewers. We strongly encourage code deposition in a community repository (e.g. GitHub). See the Nature Research guidelines for submitting code & software for further information.

## Data

Policy information about availability of data

All manuscripts must include a data availability statement. This statement should provide the following information, where applicable:

- Accession codes, unique identifiers, or web links for publicly available datasets
- A list of figures that have associated raw data
- A description of any restrictions on data availability

Data availability Statement: All SARS-CoV-2 whole genome sequences produced by NGS-SA are deposited in the GISAID sequence database and are publicly available subject to the terms and conditions of the GISAID database. The GISAID accession numbers of sequences used in the phylogenetic analysis, including

# Field-specific reporting

Please select the one below that is the best fit for your research. If you are not sure, read the appropriate sections before making your selection.

☒ Life sciences ☐ Behavioural & social sciences ☐ Ecological, evolutionary & environmental sciences

For a reference copy of the document with all sections, see nature.com/documents/nr-reporting-summary-flat.pdf

# Life sciences study design

All studies must disclose on these points even when the disclosure is negative.

| | |
|---|---|
| Sample size | No sample size calculation was performed; rather all genomic data available at the time of writing for the newly emerged Omicron variant was considered to ensure most accurate analysis and results in a timely manner. At the time of writing (11 December 2021), 553 good quality sequences of the Omicron SARS-CoV-2 variant had been produced by the NGS-SA and Botswana Harvard HIV Reference Laboratory (BHHRL) in South Africa (all fastq in SRA). We believe this was a sufficient sample size as the genomes spanned 8 of the 9 provinces of South Africa, including from multiple districts and two regions of Botswana. For phyloegentic analysis, this was analyzed against a globally representative reference set of SARS-CoV-2 genotypes (n=12 609) spanning the entire genetic diversity observed since the start of the pandemic. |
| Data exclusions | For phylogenetic analysis and time-calibrated BEAST analysis, genomes were excluded if they presented <90% coverage against the reference AND/OR have sequencing quality problem - e.g. gaps in key regions of the spike protein that causes spurious clustering. |
| Replication | Reproducibility were performed for maximum likelihood (bootstrap x1000 with FastTree) and bayesian MCMC phylogenetic tree reconstructions. We computed MCMC (Markov chain Monte Carlo) triplicate runs of 100 million states each, sampling every 10,000 steps for the Omicron dataset. All attempts at replication were successful and the MCC tree for the Omicron cluster was of high support. |
| Randomization | Experimental groups consisted of weekly batches of residual patient nasopharyngeal swabs selected for sequencing to determine the progression of weekly lineage prevalence as part of surveillance. Samples for weekly SARS-CoV-2 sequencing in South Africa and Botswana were selected at random from all relevant divisions in each country, without any clinical or geographical bias. Generally, part of the Network for Genomic Surveillance in South Africa (NGS-SA), five sequencing hubs receive randomly selected samples for sequencing every week according to approved protocols at each site. In response to a rapid resurgence of COVID-19 in Gauteng Province in November, we enriched our routine sampling with additional samples from those areas. |
| Blinding | Geographical blinding of data was not necessary for the study as it involves phylogeographical analysis. Other types of blinding were also not necessary as this was not a cohort study. |

# Reporting for specific materials, systems and methods

We require information from authors about some types of materials, experimental systems and methods used in many studies. Here, indicate whether each material, system or method listed is relevant to your study. If you are not sure if a list item applies to your research, read the appropriate section before selecting a response.

## Materials & experimental systems

| n/a | Involved in the study |
|---|---|
| ☒ | Antibodies |
| ☒ | Eukaryotic cell lines |
| ☒ | Palaeontology |
| ☒ | Animals and other organisms |
| ☐ | ☒ Human research participants |
| ☒ | Clinical data |

## Methods

| n/a | Involved in the study |
|---|---|
| ☒ | ChIP-seq |
| ☒ | Flow cytometry |
| ☒ | MRI-based neuroimaging |

## Human research participants

Policy information about studies involving human research participants

| | |
|---|---|
| Population characteristics | We obtained samples consisting of remnant nucleic acid extracts or remnant nasopharyngeal and oropharyngeal swab samples from routine diagnostic SARS-CoV-2 PCR testing from public and private laboratories in South Africa. The Omicron genomes in |

this study came from patients of ages 0-82, with an approximately equal distribution of males and females, for which the Omicron genotype was confirmed by sequencing.

Recruitment

As part of the Network for Genomic Surveillance in South Africa (NGS-SA), five sequencing hubs receive randomly selected samples for sequencing every week according to approved protocols at each site. In response to a rapid resurgence of COVID-19 in the province of Gauteng in November, we enriched our routine sampling with additional samples from this area. One bias that may be present is the ability to sequence only from the pool of patients that seek testing and that receive a positive PCR test.

Ethics oversight

The genomic surveillance in South Africa was approved by the University of KwaZulu–Natal Biomedical Research Ethics Committee (BREC/00001510/2020), the University of the Witwatersrand Human Research Ethics Committee (HREC) (M180832), Stellenbosch University HREC (N20/04/008_COVID-19), University of Cape Town HREC (383/2020), University of Pretoria HREC (H101/17) and the University of the Free State Health Sciences Research Ethics Committee (UFS-HSD2020/1860/2710). The genomic sequencing in Botswana was conducted as part of the national vaccine roll-out plan and was approved by the Health Research and Development Committee (Health Research Ethics body, HRDC#00948 and HRDC#00904). Individual participant consent was not required for the genomic surveillance. This requirement was waived by the Research Ethics Committees.

Note that full information on the approval of the study protocol must also be provided in the manuscript.

