## [Peer Review File · Nature]

Manuscript Title: Rapid epidemic expansion of the SARS-CoV-2 Omicron variant in southern Africa

Reviewer Comments & Author Rebuttals

Reviewer Reports on the Initial Version:

Referee #1 (Remarks to the Author):

This is an excellent, thorough, and clearly written manuscript presenting time-sensitive molecular, phylogenetic, and phylodynamic analyses of the omicron variant during its early period of spread (through mid December 2021). I have a few major comments that I believe could be addressed in a straightforward manner in a revision of this manuscript and hope to see this important and thorough work published soon.

Major comments (in order of appearance in the text):

Analyses of BA.2 and BA.3. Most of the presented analyses understandably focus on BA.1/omicron. However, some of the later analyses (such as the selection analyses and the recombination analyses) also include BA.2. There is no mention as to why BA.3 is not included in these analyses. My suggestion is to provide the reader with more clarity as to why related lineages BA.2 and BA.3 are not considered in some analyses but included in others. My more specific suggestion is to include more information on BA.2 and BA.3 around line 235, because these lineages presumably may also be informative for understanding the origin of BA.1. Including these related lineages in Figure 2A, and in another supplemental figure zooming in to lineages BA.1, BA.2, and BA.3 would be helpful. (I recognize that we see BA.1 and BA.2 lineages already in Extended Data Figure 5, but this is for the recombination analysis.)

Extending my comment immediately above, could you please add a supplemental figure showing the mutational profiles of BA.2 and BA.3, similar to the one shown for BA.1 in Figure 3A? Also, please provide text on why BA.3 was not examined in the recombination analysis around lines 309-316, or instead, add this analysis if there is not a good reason for not performing it. Finally, the selection analysis focuses exclusively on BA.1/omicron and BA.2. Again, add text on why BA.3 was not examined in this analysis, or instead, add this analysis if there is not a good reason for not performing it.

Interpretation of phylogenetic and phylodynamic analyses. A number of phylogenetic and phylodynamic analyses were conducted, with at times very different results. For example, lines 242-244 indicate that the time-calibrated phylogenetic analysis results in an exponential growth rate of $r = 0.136$ per day, and doubling time of 5.1 days. Then on line 247, the results when only Gauteng province sequences are considered are very different, with a faster growth rate with doubling time of only ~ 1.8 days. Finally, the birth-death phylodynamic analysis (around line 250) yields a doubling time of 3.88-3.99 days. How should we interpret this extremely wide range of estimates for doubling times in particular? Some text to interpret these disparate results would be very helpful to the reader. The epidemiological data shown in Figure 1C (top row) could also be brought in here to determine which of these disparate estimates is most likely (with a further explanation of why the other estimates differ from it).

All of the phylogenetic and phylodynamic analyses assume a constant exponential growth rate of omicron. Is this consistent with epidemiological data up through December 11, 2021 (which is I believe the latest sequence included in these analyses)?

Line 854: For the BDSKY model, the mean duration of infectiousness was fixed at 10 days. This number is based off of references 75 and 76, which estimate the duration of infectiousness for lineages that are not omicron. If you instead assume a shorter duration of infectiousness (e.g., 6 or 7 days), what happens to the effective reproductive number (R_e) estimates?

Line 876/877: Estimated R_e values of 4.17 and 3.85 for the Gauteng Province dataset (for the 3-epoch and 4-epoch BDSKY models, respectively): these R_e values were not reported in the main text, around line 250, although the phylogenetic exponential growth model's findings on this same subset of sequences was reported. To me, these estimates also seem really high. Adding some text in the main manuscript on these results and their interpretation would be helpful. Are they consistent with epidemiological data/analyses?

Figure 2B. The shown time of MRCA distribution seems inconsistent with the estimates provided in lines 242-244 (Oct 9, with 95% CI of Sept 30-Oct 20). Also, the figure legend indicates 'time of origin', when it's the MRCA time distribution that is shown, I believe, not the estimated time of origin/time of index case.

Figure 4B and related analyses: the analyses are presented in the context of level of protective immunity from a previous Delta infection. Since the first two waves in South Africa (Figure 1) were also large, some text edits (a sentence or two) would be helpful to explicitly indicate how one should think about the y-axis in Figure 4. Is this really only immune evasion from a previous Delta infection, or in some way from all previous SARS-CoV-2 infections?

Extended Data Table 3: the rate of becoming non-infectious is states as fixed at a value of 36.5/day. The caption to this table indicates that this corresponds to an infectious period of 10 days. This is not the case...a duration of infectiousness of 10 days corresponds to a rate of becoming non-infectious or 36.5 per YEAR, not per day. If the analyses were performed with a fixed rate of 36.5/DAY, then they would need to be redone. Otherwise, the text of this figure just needs to be edited.

Minor comments:

Lines 249-253: It would be helpful to state the set of sequences used for the birth-death phylodynamic analysis presented here. Also, Extended data table X needs to be added and number inserted.

Line 126: (VOC) -> (VOCs)

Line 131: all other VOC -> all other VOCs

Figure 1C, bottom row plots and figure legend: "genomic prevalence" -> consider describing what is meant by this in more detail in the figure legend, since this is a proportion?

Line 324: add 'however' in this sentence?

Line 349: haplotypes were reconstructed for this analysis, or were consensus sequences used? Also, the text in the selection section seems to 'sound' a bit different from text from the other sections in terms of how it is written. I suggest text editing to make the manuscript sound smoother in terms of voice in this section relative to the other sections.

Summary:

In all, this is a very carefully performed, well-presented analysis. I think text edits to help the reader understand and interpret the disparate phylogenetic and phylodynamic estimates better would be helpful. I also think that including a bit more analysis of related lineages BA.2 and BA.3 to help us understand the origin of omicron better would be helpful.

Katia Koelle

Referee #2 (Remarks to the Author):

Great paper. Supremely important for public health but also very interesting evolutionarily.

Referee #3 (Remarks to the Author):

Viana et al. reported the discovery, genomic and epidemiological characterisations of the Omicron variant in South Africa. This work has reveal many mutations appearing in the Omicron, and its growth advantage compared to wildtype and other variants. The authors also provided the evidence of adaptive evolution playing a significant role in the Omicron emergence. The genomic, evolutionary and epidemiological analysis were well performed and made reasonable conclusion. This scientific report provides very useful early information on the emergence of Omicron. Below are my specific comments.

1. The description of epidemiology and discovery of Omicron in the first section of the Result is very informative; Just a minor suggestion that it would help readers to understand the context even better by adding the time information about the initial epidemic waves and variants in South Africa (Line 144-146).

2. Evolutionary Origins - Line 234-238: I understand that this study focuses on BA.1; Since BA.2 and BA.3 are mentioned as the related lineage and sharing many mutations with BA.1, it would be helpful to provide slightly more information about BA.2 and BA.3, in order to better understand the evolutionary origins of Omicron. Could the authors (1) clarify whether they have included BA.2 and BA.3 in Fig. 2A the phylogenetic tree showing BA.1 having no clear basal progenitor? If so, could they be highlighted as well? (2) explain the evolutionary relationship between BA.2 & 3 and BA.1? They are totally independent lineages? Or does one branch off from the others which might give clues on the evolutionary ancestry? (3) briefly describe where and when these BA.2 & 3 were first sampled? Before or after BA.1? From South Africa too and Gauteng?

3. Line 242-243: Is the 95% credible interval referring specifically to the 95% highest posterior density?

4. Line 251-252: Extended Data Table X? Re with e subscripted?

5. Since this manuscript has no Discussion section, it would be useful for the authors to comment here at line 252-256 about the statistical confidence for the Omicron to be originated from Gauteng and spread to other provinces and Botswana.

6. Recombination Analysis - Line 309-311: Was the BA.3 also involved in the analysis?

7. Extended data table 4 (page 61): The notation for double time appears as a unknown symbol.

8. Extended data figure 5: Larger and clearer font size for the tree tip labels.

Author Rebuttals to Initial Comments:

Referee #1

General remarks: This is an excellent, thorough, and clearly written manuscript presenting time-sensitive molecular, phylogenetic, and phylodynamic analyses of the omicron variant during its early period of spread (through mid-December 2021). I have a few major comments that I believe could be addressed in a straightforward manner in a revision of this manuscript and hope to see this important and thorough work published soon.

General response: *We thank the reviewer for their thorough consideration of our work. We appreciate the encouragement and we have addressed the comments and suggestions made for improving this work. We describe below our specific responses and highlight, where relevant, our revisions to the manuscript. N.B. Line numbers refer to the clean revised version of the manuscript.*

Comment 1: Analyses of BA.2 and BA.3. Most of the presented analyses understandably focus on BA.1/omicron. However, some of the later analyses (such as the selection analyses and the recombination analyses) also include BA.2. There is no mention as to why BA.3 is not included in these analyses. My suggestion is to provide the reader with more clarity as to why related lineages BA.2 and BA.3 are not considered in some analyses but included in others.

Response 1a: *BA.1, BA.2 and BA.3 are now included in the recombination and selection analyses as detailed in responses to comments 4 and 5.*

My more specific suggestion is to include more information on BA.2 and BA.3 around line 235, because these lineages presumably may also be informative for understanding the origin of BA.1.

Response 1b: *Thank you for the suggestion. We have added more information on BA.2 and BA.3 in the text in lines 226-231: "While BA.2 and BA.3 are evolutionarily linked to*

BA.1 in that they all branch off of the same B.1.1 node without obvious progenitors, the three sub-lineages evolved independently from one another along separate branches (Extended Data Fig. 4C, 4D). The earliest specimens of BA.2 and BA.3 were both sampled after the earliest known BA.1 in South Africa (8 November 2021 at the time of writing), on 17 November 2021 in Tshwane (Gauteng) and on 18 November 2021 in a neighbouring province (North West) respectively.”

Comment 2: Including these related lineages in Figure 2A, and in another supplemental figure zooming in to lineages BA.1, BA.2, and BA.3 would be helpful. (I recognize that we see BA.1 and BA.2 lineages already in Extended Data Figure 5, but this is for the recombination analysis.)

Response 2: *As per your suggestion, we have run an updated phylogeny with additional BA.2 (n=148) and BA.3 (n=19) sequences to demonstrate the evolutionary relationship between BA.1, BA.2 and BA.3 and show the results, including a zoomed-in view of the BA.1, BA.2 and BA.3 clade in Extended Data (Extended Data Fig. 4C, 4D).*

Comment 3: Extending my comment immediately above, could you please add a supplemental figure showing the mutational profiles of BA.2 and BA.3, similar to the one shown for BA.1 in Figure 3A?

Response 3: *In the same Extended Data Figure as above, we have added mutation profiles for BA.2 and BA.3 (Extended Data Fig. 4A, 4B).*

Comment 4: Also, please provide text on why BA.3 was not examined in the recombination analysis around lines 309-316, or instead, add this analysis if there is not a good reason for not performing it.

Response 4: *BA.3 has now been included in the analysis and some trace evidence of recombination between BA.1, BA.2 and BA.3 is now found. The following changes to the*

text can be found in lines 312-323: “Potential evidence of a single recombination event involving BA.1, BA.2 and BA.3 was identified by 3SEQ ($p = 0.03$), GARD (delta c-AIC = 20) and RDP5 (GENECONV $p=0.027$; RDP $p=0.006$). within the NTD encoding region of spike. The most likely breakpoint locations for this recombination event were 21690 for the 5’ breakpoint (high likelihood interval between 15716 and 21761) and 22198 for the 3’ breakpoint (high likelihood interval between 22197 and 22774). However, these analyses could not reliably identify which of BA.1, BA.2 or BA.3 was the recombinant. Phylogenetic analysis of the genome regions bounded by these breakpoints (genome coordinates 1-21689, 21690-22198 and 22199-29903) potentially supported: (i) BA.1 having acquired the NTD encoding region of BA.3 through recombination, (ii) BA.3 having acquired the NTD encoding region of BA.1 through recombination or (iii) BA.2 having acquired the NTD encoding region of a non-BA lineage virus through recombination (**Extended Data Fig. 6**).”

Comment 5: Finally, the selection analysis focuses exclusively on BA.1/Omicron and BA.2. Again, add text on why BA.3 was not examined in this analysis, or instead, add this analysis if there is not a good reason for not performing it.

Response 5: BA.3 has now been included in the selection analysis although it should be pointed out that there are presently only sufficient BA.1 sequences available to evaluate the ongoing selection pressures acting on these viruses. The BA.2 and BA.3 sequences only really contribute to the evaluation of selection signals along the long branch(es) separating these lineages from non-BA lineage viruses.

Comment 6: Interpretation of phylogenetic and phylodynamic analyses. A number of phylogenetic and phylodynamic analyses were conducted, with at times very different results. For example, lines 242-244 indicate that the time-calibrated phylogenetic analysis results in an exponential growth rate of $r = 0.136$ per day, and doubling time of 5.1 days. Then on line 247, the results when only Gauteng province sequences are considered are

very different, with a faster growth rate with doubling time of only ~1.8 days. Finally, the birth-death phylodynamic analysis (around line 250) yields a doubling time of 3.88-3.99 days. How should we interpret this extremely wide range of estimates for doubling times in particular? Some text to interpret these disparate results would be very helpful to the reader. The epidemiological data shown in Figure 1C (top row) could also be brought in here to determine which of these disparate estimates is most likely (with a further explanation of why the other estimates differ from it).

Response 6: *The estimated doubling times are not so different once the credible intervals of the estimates are considered. Further, the birth-death skyline analyses have been re-calculated using exactly the same alignment as for the coalescent analyses, with the effect that the growth rates estimated using the two methods are now more similar to each other (see response 9 below). The BDSKY analyses have been re-computed, using a larger complete data set that is identical to the data set used in the coalescent analyses, facilitating direct comparison between estimates. The estimated R_e for the Gauteng Province dataset is now 3.86 (95% CI 3.43 - 4.29) and 3.61 (95% CI 3.20 - 4.02) for the 3-epoch and 4-epoch model respectively. These values are lower, more reasonable and more consistent with other epidemiological estimates. Further, the data sets represent different geographic regions, hence differences between the data sets are to be expected given spatial heterogeneity in transmission intensity, and the effects of spatial structure on phylogenetic topology.*

Comment 7: All of the phylogenetic and phylodynamic analyses assume a constant exponential growth rate of omicron. Is this consistent with epidemiological data up through December 11, 2021 (which is I believe the latest sequence included in these analyses)?

Response 7: *Yes this is consistent with the epidemiological data, demonstrating exponential growth of cases and estimated effective reproduction number (R_e) >1 through to at least 17 December (see R_e estimates from the National Institute of*

Communicable Diseases at <https://www.nicd.ac.za/diseases-a-z-index/disease-index-covid-19/surveillance-reports/covid-19-special-reports/the-initial-and-daily-covid-19-effective-reproductive-number-in-south-africa/>)

Comment 8: Line 854: For the BDSKY model, the mean duration of infectiousness was fixed at 10 days. This number is based off of references 75 and 76, which estimate the duration of infectiousness for lineages that are not omicron. If you instead assume a shorter duration of infectiousness (e.g., 6 or 7 days), what happens to the effective reproductive number (R_e) estimates?

Response 8: *We have re-run the analyses of the full dataset using fixed average durations of infectiousness of 7 and 14 days. These changes both leave the TMRCA largely unchanged. Increasing the average duration of infectiousness leads to larger estimates of R_e (and smaller sampling proportions) while decreasing it leads to smaller estimates of R_e (and larger sampling proportions). A more thorough description of this sensitivity analysis is included in Methods (lines 874-895).*

Comment 9: Line 876/877: Estimated R_e values of 4.17 and 3.85 for the Gauteng Province dataset (for the 3-epoch and 4-epoch BDSKY models, respectively): these R_e values were not reported in the main text, around line 250, although the phylogenetic exponential growth model's findings on this same subset of sequences was reported. To me, these estimates also seem really high. Adding some text in the main manuscript on these results and their interpretation would be helpful. Are they consistent with epidemiological data/analyses?

Response 9: *The BDSKY analyses have been re-computed, using a larger complete data set that is identical to the data set used in the coalescent analyses, facilitating direct comparison between estimates. The estimated R_e for the Gauteng Province dataset is now 3.86 (95% CI 3.43 - 4.29) and 3.61 (95% CI 3.20 - 4.02) for the 3-epoch and 4-epoch model respectively. These values are lower, more reasonable and more*

consistent with other epidemiological estimates. Possible reasons for this drop are that (i) the previous, smaller data set was less reliable, or (ii) a less informative prior that allowed for a higher sampling proportion was used for the Gauteng analysis (a higher sampling proportion compared to South Africa overall), in accordance with observations from comparisons between genome count and case count over the study period for Gauteng.

Comment 10: Figure 2B. The shown time of MRCA distribution seems inconsistent with the estimates provided in lines 242-244 (Oct 9, with 95% CI of Sept 30-Oct 20). Also, the figure legend indicates ‘time of origin’, when it’s the MRCA time distribution that is shown, I believe, not the estimated time of origin/time of index case.

Response 10: *The figure legend stating ‘time of origin’ was an error and has now been corrected to say MRCA time. Thank you for pointing this out.*

Comment 11: Figure 4B and related analyses: the analyses are presented in the context of level of protective immunity from a previous Delta infection. Since the first two waves in South Africa (Figure 1) were also large, some text edits (a sentence or two) would be helpful to explicitly indicate how one should think about the y-axis in Figure 4. Is this really only immune evasion from a previous Delta infection, or in some way from all previous SARS-CoV-2 infections?

Response 11: *We now better clarify the definition of population immunity in the main text and caption of Figure 4. Specifically, in lines 398-401: “... population immunity against infection by, and transmission of, the competing variant Delta that is afforded by prior infections with wild-type, Beta, Delta, and other strains during the three previous epidemic waves in South Africa, and/or vaccination.”*

Comment 12: Extended Data Table 3: the rate of becoming non-infectious is stated as fixed at a value of 36.5/day. The caption to this table indicates that this corresponds to an infectious period of 10 days. This is not the case...a duration of infectiousness of 10 days

corresponds to a rate of becoming non-infectious or 36.5 per YEAR, not per day. If the analyses were performed with a fixed rate of 36.5/DAY, then they would need to be redone. Otherwise, the text of this figure just needs to be edited.

Response 12: *The reviewer is correct – thanks for pointing out this error. This typo has been corrected in Extended Data Table 3. No further analyses are needed.*

Comment 13: Lines 249-253: It would be helpful to state the set of sequences used for the birth-death phylodynamic analysis presented here. Also, Extended data table X needs to be added and number inserted.

Response 13: *The BDSKY analyses were repeated using the same alignment of genomes used for the coalescent analyses, excluding the one BA.2 genome from South Africa (n=552 BA.1 assigned genomes from South Africa and Botswana and n=277 genomes from Gauteng province only). This is now stated in the methods (lines 833-835) and in the main text (lines 243-248).*

Comment 14: Line 126: (VOC) -> (VOCs)

Response 14: *Changed as suggested*

Comment 15: Line 131: all other VOC -> all other VOCs

Response 15: *Changed as suggested*

Comment 16: Figure 1C, bottom row plots and figure legend: “genomic prevalence” -> consider describing what is meant by this in more detail in the figure legend, since this is a proportion?

Response 16: *Thanks for the suggestions, a sentence was added to the Figure 1C legend to explain this, as follows: “Genomic prevalence here is equivalent to the proportion of weekly surveillance sequences genotyped as being Omicron.”*

Comment 17: Line 324: add ‘however’ in this sentence?

Response 17: *Added as suggested*

Comment 18: Line 349: haplotypes were reconstructed for this analysis, or were consensus sequences used?

Response 18: *Consensus sequences were used. The word “haplotype” here means “genetically distinct genome sequences.” This is now more clearly stated in the main text.*

Comment 19: Also, the text in the selection section seems to ‘sound’ a bit different from text from the other sections in terms of how it is written. I suggest text editing to make the manuscript sound smoother in terms of voice in this section relative to the other sections.

Response 19: *We have tried to make the language more consistent in our revision*

Referee #3

General comment: Viana et al. reported the discovery, genomic and epidemiological characterisations of the Omicron variant in South Africa. This work has reveal many mutations appearing in the Omicron, and its growth advantage compared to wildtype and other variants. The authors also provided the evidence of adaptive evolution playing a significant role in the Omicron emergence. The genomic, evolutionary and epidemiological analysis were well performed and made reasonable conclusion. This scientific report provides very useful early information on the emergence of Omicron. Below are my specific comments.

General response: *We thank the reviewer for their thorough consideration of our work. We appreciate the encouragement and we have addressed the comments and suggestions made for improving this work. We describe below our specific responses and highlight, where relevant, our revisions to the manuscript.*

Comment 1: The description of epidemiology and discovery of Omicron in the first section of the Result is very informative; Just a minor suggestion that it would help readers to understand the context even better by adding the time information about the initial epidemic waves and variants in South Africa (Line 144-146).

Response 1: *Thank you for this suggestion. We have added this information in lines 136-140: “The three distinct epidemic waves of SARS-CoV-2 experienced by southern African countries were each driven by different variants: the first between June and August 2020 by descendants of the B.1 lineage, the second between November 2020 and February 2021 by the Beta VOC, and the third between May and September 2021 by the Delta VOC...”*

Comment 2: Evolutionary Origins - Line 234-238: I understand that this study focuses on BA.1; Since BA.2 and BA.3 are mentioned as the related lineage and sharing many

mutations with BA.1, it would be helpful to provide slightly more information about BA.2 and BA.3, in order to better understand the evolutionary origins of Omicron. Could the authors (1) clarify whether they have included BA.2 and BA.3 in Fig. 2A the phylogenetic tree showing BA.1 having no clear basal progenitor? If so, could they be highlighted as well? (2) explain the evolutionary relationship between BA.2 & 3 and BA.1? They are totally independent lineages? Or does one branch off from the others which might give clues on the evolutionary ancestry? (3) briefly describe where and when these BA.2 & 3 were first sampled? Before or after BA.1? From South Africa too and Gauteng?

Response 2: *BA.3 was not initially included in the phylogeny as it had not yet been designated at the time of writing, and only a few initial BA.2 sequences were included (n=4). To address these questions, we have now run an updated phylogeny with additional BA.2 (n=148) and BA.3 (n=19) sequences to demonstrate the evolutionary relationship between BA.1, BA.2 and BA.3. We show the results, including a zoomed-in view of the BA.1, BA.2 and BA.3 clade in Extended Data Fig. 4C, 4D. We have also added additional text to the manuscript in lines 226-231: “While BA.2 and BA.3 are evolutionarily linked to BA.1 in that they all branch off of the same B.1.1 node without obvious progenitors, the three sub-lineages evolved independently from one another along separate branches (**Extended Data Fig. 4C, 4D**). The earliest specimens of BA.2 and BA.3 were both sampled after the earliest known BA.1 in South Africa (8 November 2021 at the time of writing), on 17 November 2021 in Tshwane (Gauteng) and on 18 November 2021 in a neighbouring province (North West) respectively.”*

Comment 3: Line 242-243: Is the 95% credible interval referring specifically to the 95% highest posterior density?

Response 3: *Yes correct these are 95% highest posterior density intervals. We have now changed in the relevant text.*

Comment 4: Line 251-252: Extended Data Table X? Re with e subscripted?

Response 4: *Thanks for pointing out these typos/ommissions. We have now corrected.*

Comment 5: Since this manuscript has no Discussion section, it would be useful for the authors to comment here at line 252-256 about the statistical confidence for the Omicron to be originated from Gauteng and spread to other provinces and Botswana.

Response 5: *To address this point, we have added the following sentence in lines 251-253: “However, this does not imply that Omicron originated in Gauteng and these phylogeographic inferences could change as further genomic data accumulates from other locations.”*

Comment 6: Recombination Analysis - Line 309-311: Was the BA.3 also involved in the analysis?

Response 6: *BA.3 is now included in the updated analysis*

Comment 7: Extended data table 4 (page 61): The notation for double time appears as a unknown symbol.

Response 7: *Thanks – this has now been corrected.*

Comment 8: Extended data figure 5: Larger and clearer font size for the tree tip labels.

Response 8: *Thanks - the new version of this figure (now Extended Data Figure 6) shows the important information in clearer manner with larger font size. However, because of the number of sequences, it is not possible to increase font size for all tip labels. The figure file can be zoomed-in to display all information.*